# Multiple cAMP/PKA complexes at the STIM1 ER/PM junction specified by E-Syt1 and E-Syt2 reciprocally gates ANO1 (TMEM16A) via Ca²⁺

Wei-Yin Lin[1], Woo Young Chung [1], Seonghee Park[2], Ava Movahed Abtahi [1], Benjamin Leblanc[1], Malini Ahuja[1] & Shmuel Muallem [1] ✉

ANO1 plays a crucial role in determining numerous physiological functions, including epithelial secretion, yet its regulatory mechanisms remain incompletely understood. Here, we describe a fundamental dynamic regulation of ANO1 surface expression and Ca²⁺-dependent gating via the cAMP/PKA pathway at the STIM1 ER/PM junctions. At these junctions, STIM1 assembles AC-AKAP-PKA complexes, while E-Syt1 mediates formation of ANO1-VAPA-IRBIT-E-Syt1-AC8-AKAP5-PKA complex, that phosphorylates ANO1 S673, increasing ANO1 Ca²⁺ affinity. Within these complexes, the Ca²⁺ and cAMP pathways act synergistically to enhance ANO1 function. By contrast, E-Syt2 dissociates the ANO1-VAPA interaction, forming ANO1-IRBIT-E-Syt2-AC6-AKAP11-PKA complex that phosphorylates ANO1 S221, which markedly reduces ANO1 Ca²⁺ affinity. The effects of the E-Syts are primarily mediated by their reciprocal regulation of junctional PI(4)P, PI(4,5)P₂ and PtdSer. Accordingly, IRBIT deletion in mice impairs receptor-stimulated activation of ANO1 and fluid secretion. These findings should have broad implications for ANO1 roles and functions across various tissues.

Anoctamin 1 (ANO1, TMEM16A) is the primary, nearly ubiquitously expressed, Ca²⁺-activated Cl⁻ channel with numerous physiological roles[1]. Among others, ANO1 controls the neuronal action potential to affect pain transduction[2], muscle contractility[3] and cell proliferation in cancer[4]. In epithelial tissues, such as the salivary glands[5,6], the pancreas[7], and the airway[3,8], ANO1 drives epithelial fluid and electrolyte secretion[9]. Accordingly, understanding the function and regulation of ANO1 is essential for understanding epithelial function in health and disease.

ANO1 is directly activated by Ca²⁺ with two Ca²⁺ ions interacting with the orthostatic Ca²⁺ binding site formed by residues of the transmembrane domains (TMD) α6 and α7[1,10–12] and is regulated by the interaction of Ca²⁺ with an allosteric Ca²⁺ binding site formed by

residues of TMD α2 and α10[13]. A well-established mode of ANO1 regulation is by the lipid PI(4,5)P₂, which appears to interact with 4 separate sites[14,15]. The molecular mechanism by which PI(4,5)P₂ regulates the channel function is not clear, although PI(4,5)P₂ binding to site 4, which is formed by TMD α6 and α7 close to the Ca²⁺ binding site, partially inactivates the channel[16]. Phosphorylation of S673 in the mouse ANO1 (ac) isoform by CAMKII modulates this form of regulation[16]. Effects of other phosphorylation sites, in particular by conserved PKA sites, have not been studied, let alone in the context of synergy between the Ca²⁺ and cAMP signaling pathways that regulate many physiological responses[17–21].

The classical Ca²⁺/cAMP signaling synergy determines epithelial fluid and electrolyte secretion, which is regulated by the combined

---

[1]The Epithelial Signaling and Transport Section and The National Institute of Dental and Craniofacial Research, National Institutes of Health, Bethesda, MD, USA. [2]Department of Physiology, Ewha Womans University College of Medicine, Seoul, Korea. ✉e-mail: shmuel.muallem@nih.gov

**Fig. 1 | Deletion of IRBIT in mice increases receptor-stimulated Ca²⁺ signaling but inhibits activation of ANO1 and fluid secretion.** A–D Parotid acini from 3 wild-type and 3 IRBIT⁻/⁻ mice were used to measure Ca²⁺ in response to 10 nM (**A**, **B**) or 100 μM carbachol (**C**, **D**) in isolated acinar clusters. Ca²⁺ oscillations (example traces in **A**) were analyzed in 4 experiments and 15 wild-type and 4 experiments, and 19 IRBIT⁻/⁻ acinar clusters to obtain the number of cells showing Ca²⁺ oscillations and total amount of Ca²⁺ increase (**B**). For Ca²⁺ release and influx (**C**, **D**), the acini were stimulated first in Ca²⁺-free media, and then Ca²⁺ was added back. Average traces with acini from one mouse are shown in (**C**), and the Ca²⁺ release and influx in 30 acinar clusters from 3 mice of each line are shown in (**D**). **E** Saliva secretion was stimulated by 0.125 mg/kg pilocarpine in 4 wild-type and 3 IRBIT⁻/⁻ mice. **F**, **G** Example traces of intracellular Cl⁻ measurements with the Cl⁻ dye MQAE and determined as F/F₀ in wild-type acini treated with vehicle (black trace) or 5 μM Ani9 (red trace) and stimulated with 100 μM carbachol (**F**) or acini from wild-type (black/green traces) or IRBIT⁻/⁻ acini (blue, red traces) stimulated with 100 μM carbachol (**G**). (**H**–**J**), Dose-response for carbachol-stimulated ANO1 activity in acini from wild-type (black) or IRBIT⁻/⁻ mice (red) (**H**). Results from 6 independent experiments were fitted to a Michaelis-Menten equation to obtain the Vmax (**I**) and Apparent Km for carbachol (**J**). All results are shown as mean ± s.e.m.

action of parasympathetic and sympathetic inputs that act through the Ca²⁺ and cAMP second messenger systems, respectively[9,17,22–24]. Previously, we have shown that the scaffold protein IRBIT (IP₃ receptor binding protein released with IP₃) plays a role in the synergy between the Ca²⁺ and cAMP systems[17]. IRBIT was discovered as a binding partner of the IP₃Rs that inhibits the channel to prevent uncontrolled Ca²⁺ release[25,26]. The specific plasma membrane (PM) domain at which Ca²⁺ and cAMP synergy takes place, the molecular mechanism by which it occurs, and the physiological significance of the interaction at the restricted cellular domain are not known.

The present studies started with a paradoxical observation; deletion of the IP₃Rs inhibitor IRBIT in the salivary glands and the pancreas increased the receptor-evoked Ca²⁺ signaling as expected, yet it markedly inhibited ANO1 activity. To solve this conundrum, we noticed that IRBIT and ANO1 possess a motif recognized by the VAMP-associated proteins (VAPA and VAPB) that recruit proteins to membrane contact sites (MCS)[27–29], including the ER/PM junctions[30,31]. We discovered that VAPA (but not VAPB) and IRBIT complex with ANO1, and target ANO1 to the restricted and specific STIM1-formed ER/PM junctions. At the junctions, ANO1 interacts with separate cAMP signaling complexes that reciprocally regulate ANO1 gating by Ca²⁺. Thus, VAPA and E-Syt1 orchestrate the formation of ANO1-VAPA-IRBIT-E-Syt1-AC8-AKAP5-PKA that phosphorylates S673 to facilitate the activation of ANO1. Conversely, E-Syt2 dissociates the VAPA-ANO1 complex and promotes the assembly of the ANO1-IRBIT-E-Syt2-AC6-AKAP11-PKA complex that phosphorylates S221 to hinder activation of ANO1 by Ca²⁺. The complexes act by two mechanisms. Firstly, they affect the surface expression of ANO1. Secondly, the VAPA-IRBIT-E-Syt1-AC8-AKAP5-PKA complex increase the affinity of ANO1 for Ca²⁺,

while the IRBIT-E-Syt2-AC6-AKAP11-PKA markedly reduces the affinity of ANO1 for activation by Ca²⁺. Disruption of the ER/PM junctions by deletion of STIM1 eliminated all forms of regulation of ANO1 by the cAMP/PKA pathway. Remarkably, mutating the lipid transfer function of the SMP domains reversed the function of the E-Syts. Depletion of native VAPA, IRBIT, the E-Syts, the relevant ACs, and AKAPs revealed the operation of these complexes with the native ANO1 in a human salivary gland acinar cell line. These findings unveil an intricate regulatory mechanism of ANO1 by the cAMP/PKA pathway, facilitated by VAPA-IRBIT- mediated assembly and synergy of the Ca²⁺ and cAMP signaling pathways. The synergy between the signaling pathways takes place exclusively at the STIM1 formed ER/PM junctions, expanding the diverse modes of regulation governing the Cl⁻ channel function of ANO1 with broad physiological implications.

## Results

### Paradoxical effect of IRBIT on receptor-stimulated Ca²⁺ signaling, Ca²⁺-activated ANO1 function and salivary fluid secretion

IRBIT competes with IP₃ for binding to the IP₃Rs and inhibits Ca²⁺ release[26]. At the resting state, IRBIT is bound to the IP₃Rs, and cell stimulation generates IP₃ that binds to the IP₃Rs to dissociate IRBIT[17] and release Ca²⁺ from the ER[26]. Previous studies reported the role of IRBIT in Ca²⁺ signaling in cell lines. We compared the receptor-evoked Ca²⁺ signals in acinar cells obtained from the parotid glands of wild-type and IRBIT⁻/⁻ mice (Fig. 1A–D). Physiological stimulation of muscarinic receptors resulted in Ca²⁺ oscillations, and deletion of IRBIT increased the number of responding cells and the strength of the oscillations (Fig. 1A, B). However, intense stimulation of the same receptors that generate very high IP₃ levels revealed no difference in

the amount of $Ca^{2+}$ released from the ER in the absence of extremal $Ca^{2+}$ or the $Ca^{2+}$ influx measured by re-addition of external $Ca^{2+}$ (Fig. 1C, D).

Since we are interested in epithelial fluid secretion, and salivation is a $Ca^{2+}$-stimulated activity[9], we compared salivation in wild-type and IRBIT[−/−] mice stimulated with a low concentration of pilocarpine. Surprisingly, deletion of IRBIT that increased $Ca^{2+}$ signaling resulted in reduced rather than the expected increase in salivation (Fig. 1E). $Cl^-$ flux by ANO1 drives fluid secretion by salivary acinar cells[5] and deletion of ANO1 markedly impairs salivation[5,32]. Therefore, we measured the effect of IRBIT deletion on ANO1-mediated $Cl^-$ flux with the $Cl^-$ sensitive dye MQAE[33]. The activity of ANO1 was assessed using the specific ANO1 inhibitor Ani9[34]. Figure 1F shows that replacing medium $Cl^-$ with $NO_3^-$ resulted in a slow $Cl^-/NO_3^-$ exchange, the rate of which was markedly increased by stimulating the acinar cells with 100 μM carbachol. Nearly complete inhibition of this flux by 5 μM Ani9 indicates that carbachol stimulation prominently activated ANO1 in parotid acinar cells, in agreement with previous studies[5]. Similar measurements with IRBIT[−/−] cells showed a marked reduction in ANO1 activity (Fig. 1G), although the $Ca^{2+}$ signal was similar under this stimulation paradigm (Fig. 1C, D). Measurement of the concentration dependence in Fig. 1H–J revealed that deletion of IRBIT reduced both the maximal response and the apparent affinity for receptor stimulation. Thus, the deletion of IRBIT that increased the $Ca^{2+}$ signal, paradoxically reduced activation of the $Ca^{2+}$-activated $Cl^-$ channel ANO1, resulting in impaired epithelial fluid secretion and salivation. The studies described below attempt to solve this conundrum.

## IRBIT mediates synergistic activation of ANO1 by the $Ca^{2+}$ and the PKA signaling pathway

To understand the regulation of ANO1 by IRBIT, we turned to a model system that enabled specific protein expression and knockdown. Measurement of ANO1 current in HEKT cells transfected with high levels of ANO1 showed activation of the current by $Ca^{2+}$, inhibition by Ani9, and regulation of voltage dependence by $Ca^{2+}$. The current was increased by IRBIT, but stimulation with 10 μM forskolin and 100 μM of the phosphodiesterase inhibitor IBMX (F/I) was required to see an effect of the cAMP/PKA pathway (Supplementary Fig. 1A, B). However, at lower ANO1 expression levels, IRBIT similarly increased the current density and stimulation with 1 μM forskolin and 10 μM IBMX was sufficient to prominently enhanced the current (Supplementary Fig. 1C). This may be due to a need for an optimal ANO1/cAMP pathway ratio and excessive stimulation of the PKA pathway. Therefore, all subsequent experiments were conducted under conditions of lower ANO1 expression, ensuring that the measured currents more closely resembled native ANO1 activity (see below). Increasing $Ca^{2+}$ from 0.1 to 1 μM enhanced the effect of cAMP alone but minimized the effect of cAMP when measured in the presence of IRBIT (Supplementary Fig. 1C). IRBIT increased the current even when pipette $Ca^{2+}$ was buffered with the fast $Ca^{2+}$ buffer 5 mM BAPTA that eliminated the effect of cAMP (Supplementary Fig. 1D). Most experiments used solutions with a symmetrical $Cl^-$ concentration of 140 mM. Reducing the cytoplasmic $Cl^-$ to 40 mM had no effect on ANO1 currents in the presence of IRBIT and F/I (Supplementary Fig. 1E).

Next, we measured the $Ca^{2+}$ dependence of ANO1 activation (Fig. 2A–C). In this and all other $Ca^{2+}$-dependence of ANO1 activation, to obtain the $Ca^{2+}$-dependent current, the current measured in the presence of 5 mM EGTA, and no added $Ca^{2+}$ was subtracted (total current is in Supplementary Fig. 1F–H). IRBIT increased the current Vmax independent of the cAMP pathway but increased the apparent affinity (app-Km) for activation of ANO1 by $Ca^{2+}$ that was dependent on the cAMP pathway. An increase in Vmax suggests that IRBIT enhances ANO1 surface expression and/or increase the channel open probability (NPo). We used two approaches to test this: First, we measured surface ANO1 by biotinylation, and Fig. 2D shows that IRBIT increased surface

ANO1. Second, we evaluated surface ANO1 by confocal and TIRF microscopy. Confocal imaging showed that ANO1 is primarily localized at the plasma membrane (PM), with some proteins in close proximity to the PM (Supplementary Fig. 1I). TIRF imaging, which reports proteins at the cell surface and within 100–200 nm of the PM of live cells, showed a punctate pattern of ANO1 expression at the TIRF plane (Supplementary Fig. 1J). The summary of the TIRF measurements in Fig. 2E revealed that IRBIT and F/I stimulation increased ANO1 at the TIRF plane, consistent with findings from the biotinylation assay. Consistency between biotinylation and TIRF measurements were observed in additional experiments (see below). Since both current and TIRF measurements were in live cells, we used TIRF measurements in several experiments to evaluate changes in ANO1 levels at the TIRF field as an indication of changes in surface expression. Finally, we used FRET to measure the association between ANO1 and IRBIT, which is required for the effects of IRBIT. The FRET between ANO1-YFP and IRBIT-CFP was significantly increased by elevating cytoplasmic cAMP and by increasing cytoplasmic $Ca^{2+}$ by treatment with the SERCA pump inhibitor CPA (Supplementary Fig. 1K).

We used several phosphoproteomic repeats with cells expressing ANO1 and ANO1 + IRBIT and treated with vehicle or 10 μM forskolin and 1 mM IBMX in attempts to identify the residues phosphorylated by PKA that affect ANO1 current when regulated by IRBIT (Fig. 2A). In all conditions, the only residue identified is S106, the same residue reported in a comprehensive analysis of cellular PKA sites[35]. Since this residue did not exclusively mediate the cAMP effect (Supplementary Fig. 2A, B), we used the GPS-PKA prediction tool (https://gps.biocuckoo.cn/online_LGBM.php) to predict the ANO1 PKA sites (Supplementary Fig. 2A). Since the experiments in Fig. 2A–C used pipette solutions lacking ATP to avoid premature and uncontrolled phosphorylation, we tested whether including ATP in the pipette solution would affect regulation by IRBIT and F/I (Fig. 2F). The presence of ATP in the pipette solution slightly increased current density but did not affect regulation by IRBIT or F/I. Since the compartmentalized ATP was sufficient for the regulation by F/I, ATP was not included in the pipette solution for the remaining experiments. Mutating the predicted PKA sites and tested for activation by IRBIT and stimulation by cAMP eliminated activation of ANO1 by IRBIT and cAMP (Supplementary Fig. 2B). Although interesting, these mutants were not analyzed further as part of the present studies. However, the ANO1(S673A) mutant responded normally to IRBIT but eliminated the effect of cAMP, indicating that ANO1(S673A) lost response to cAMP stimulation. Accordingly, the current of ANO1(S673A) + IRBIT was similar to that measured with ANO1 + IRBIT treated with 100 nM of the specific PKA inhibitor H89 (Fig. 2G). By contrast, the current by the phosphomimetic ANO1(S673D) treated with IRBIT was the same as that of ANO1 + IRBIT +F/I and did not respond to further stimulation with F/I (Fig. 2F). The location of S673 relative to the ANO1 $Ca^{2+}$ binding site is shown in Supplementary Fig. 2C, D. Finally, part of the effects of IRBIT and F/I stimulation on ANO1 and of the ANO1 mutants' currents were due to their effect on ANO1 levels at the TIRF field and, consequently, its presence on the cell surface (Fig. 2H). Notably, if phosphorylation of S673 mediates the effect of IRBIT + PKA, the phosphomimetic ANO1(S673D) is predicted to increase the channel Vmax and apparent affinity for $Ca^{2+}$. The results in Fig. 2I–K (total current in Supplementary Fig. 2E–G) confirm this prediction with ANO1(S673D) increasing the Vmax from 11.8 to 19.8 pA/pF and reducing app-Km from 0.695 to 0.273 μM $Ca^{2+}$. Furthermore, stimulation with F/I had no additional impact on the Vmax and app-Km of ANO1(S673A).

## Regulation of ANO1 by IRBIT and PKA takes place at the ER/PM junction; Role of VAPA

To explore the mechanism and requirement for regulation of ANO1 by PKA we asked whether the regulation is affected by PM $PI(4,5)P_2$ level. Previous studies have reported that depletion of PM $PI(4,5)P_2$ partially

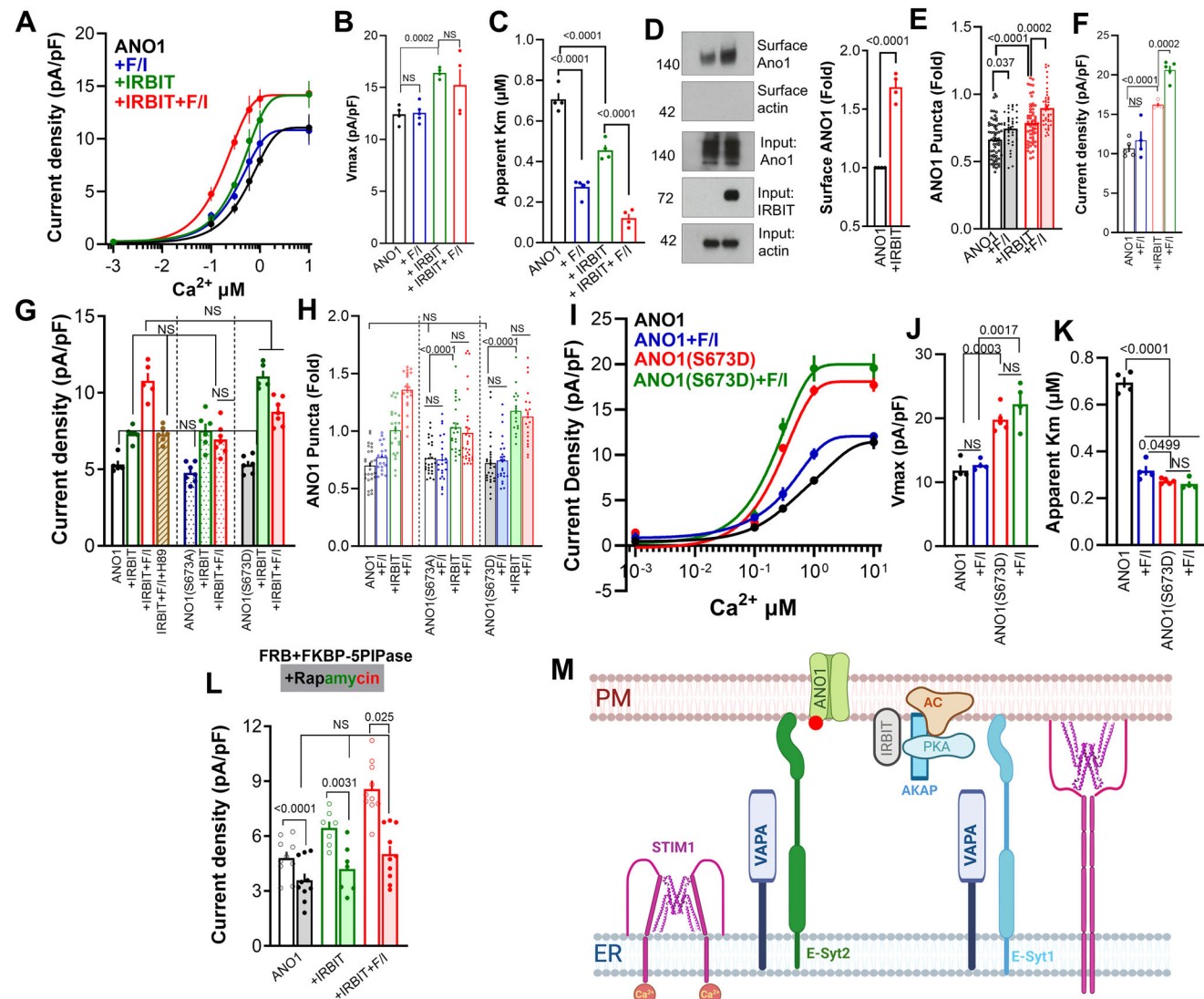

**Fig. 2 | Roles of IRBIT in regulation by the cAMP/PKA pathway of ANO1 activation by Ca²⁺ and surface expression.** A–C Shown are the Ca²⁺ dependence of ANO1 (black, blue), in the presence of IRBIT (green, red) and F/I stimulation (blue, red) (**A**), the Vmax (**B**) and apparent Km for Ca²⁺ (**C**) from 4 independent experiments. **D** Surface expression of ANO1 in cells transfected with vector or IRBIT (red) was assayed by biotinylation, with example blots and the average of 4 independent experiments. **E** Analysis of surface ANO1 by TIRF in 6 independent experiments in the presence and absence of IRBIT and F/I. **F** Current was measured in 4 independent experiments with a pipette solution containing 1 mM ATP and 0.3 μM Ca²⁺. **G** Current was measured in 5 independent experiments in cells expressing ANO1 (black) and IRBIT (green) and stimulated with F/I (red); ANO1(S673A), IRBIT and stimulated with F/I (dotted columns); ANO1(S673D), IRBIT and stimulated with F/I (filled columns). H89 0.1 μM (brown) was included in the pipette solution. **H** ANO1

levels were measured by TIRF in 4 independent experiments in cells expressing ANO1, ANO1(S673A), or ANO1(S673D) as indicated, and empty vector (black), IRBIT (green) and stimulated with F/I (Blue, red). **I–K** Shown are the Ca²⁺ dependence of ANO1 and ANO1(S673D) stimulated with F/I (blue, green) (**I**), the Vmax (**I**) and Km for Ca²⁺ (**J**) from 4 independent experiments. **L** FRB+5PIPase-FKBP and rapamycin were used to deplete PI(4,5)P₂ and measure ANO1 current in the presence of IRBIT and F/I. **M** A Schematic illustrating the ANO1-associated tether and cAMP/PKA pathway proteins that form regulatory complexes at the ER/PM junctions, as examined in the present studies, is shown. In (**G**) pipette solution Ca²⁺ was 0.3 μM. In these and other experiments below, unless otherwise indicated, experiments were with pipette solution in which Ca²⁺ was buffered to 0.3 μM. The 0.3 μM Ca²⁺ was selected since the Ca²⁺-dependent current can be clearly resolved, and the effect of the various regulators was optimal. All results are shown as mean ± s.e.m.

---

inhibited the current of the ANO1(ac) isoform[16]. We used the FRB/FKBP/rapamycin system[36] to acutely deplete PI(4,5)P₂[37,38] which slightly reduced ANO1 current but eliminated the effect of IRBIT and F/I (Fig. 2L). PI(4,5)P₂ is concentrated at the MCS site ER/PM junctions[39,40], suggesting that IRBIT and the cAMP signaling pathway communicate with ANO1 at the ER/PM junctions. The complexes that ANO1 may form with components of the cAMP pathway at the ER/PM junctions, and key junctional tethers that regulate these complexes are modeled in Fig. 2M and are examined below.

IRBIT does not have a membrane-spanning domain. However, analysis of the IRBIT sequence revealed the potential VAP (VAMP-associated protein) binding FFAT motif[73] **DSYSSAASY**TD**S**SDDE[88]. This

led us to examine the role of VAP complexes in the regulation of ANO1, as depicted in the right-side complex in Fig. 2M. We mutated the underlined residues that are predicted to disrupt the IRBIT VAP recognition motif[29,41], thereby eliminating the effect of IRBIT on ANO1 current (Supplementary Fig. 3A). There are two VAP isoforms, VAPA and VAPB, that are thought to have redundant functions[41,42]. However, measuring their interaction with ANO1 by FRET showed a strong interaction between ANO1 and VAPA, which was markedly reduced upon mutation of the ANO1 FFAT motif. By contrast, VAPB showed the same FRET signal with ANO1 and the ANO1 FFAT mutant (Supplementary Fig. 3B, C). Further, measuring the effect of the two VAP isoforms on ANO1 current showed that VAPA prominently activated

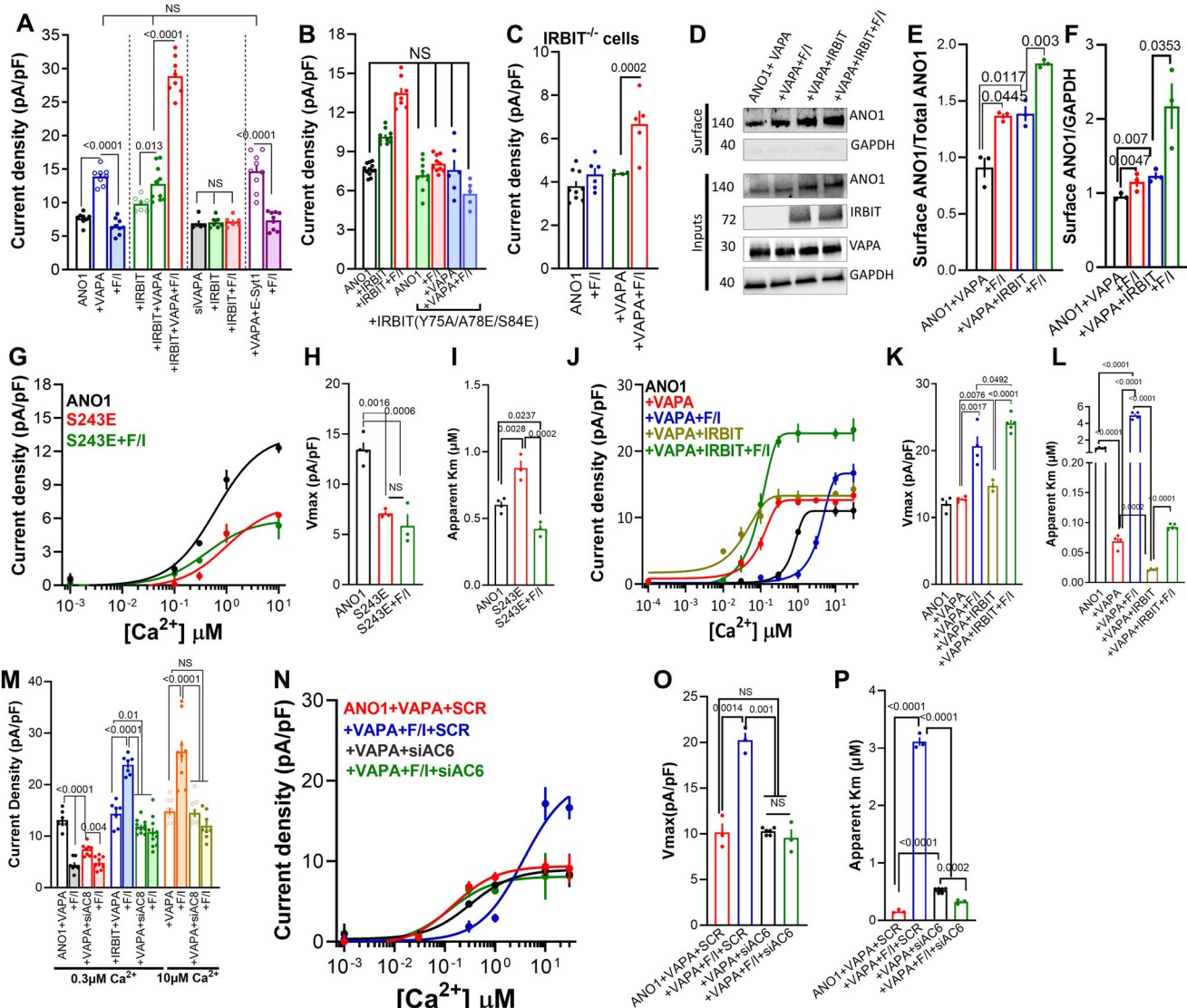

**Fig. 3 | Roles of VAPA in regulation by the cAMP/PKA pathway of ANO1 activation by $Ca^{2+}$ and surface expression.** **A** ANO1 current was measured in 4 independent experiments in the presence of VAPA (blue) and F/I, IRBIT, IRBIT + VAPA and stimulated with F/I as indicated in the columns; Cells treated with siVAPA (filled columns), IRBIT, and stimulated with F/I; Celle expressing VAPA + E-Syt1 and stimulated with F/I. **B** Current was measured in cells transfected with ANO1 (black) and either IRBIT (open columns) or IRBIT(Y75A/A78E/S84E) FFAT site mutant (filled columns) and VAPA (blue and light blue) and stimulated with F/I. The results are from 4 independent experiments. **C** ANO1 was expressed in IRBIT$^{-/-}$ cells alone (black) and with VAPA (green) and stimulated with F/I (blue, red). The results are from 3 independent experiments. **D**–**F** Effect of VAPA (black) and VAPA + IRBIT (blue) before and after treatment with F/I (red, green) on ANO1 surface expression that was measured by biotinylation in 3 independent experiments. Example blots are in (**D**) and averages relative to total ANO1 and to GAPDH are in (**E**) and (**F**),

respectively. **G**–**I** The $Ca^{2+}$ dependence of the ANO1(black) and ANO1(S243E) (red) stimulated with F/I (green) (**G**), the Vmax (**H**) and apparent Km for $Ca^{2+}$ (**I**) were obtained in 3 independent experiments. **J**–**L** The $Ca^{2+}$ dependence of the ANO1 in the presence of vector (black), VAPA (red) and stimulated with F/I (blue), VAPA + IRBIT (dark yellow), and stimulated with F/I (green) (**J**), the Vmax (**K**) and apparent Km for $Ca^{2+}$ (**L**) were obtained from 4 independent experiments. **M** Current was measured in pipette solution with $Ca^{2+}$ buffered to 0.3 (5 independent experiments) or 10 μM (4 independent experiments) in cells transfected with ANO1 and VAPA (black, red, orange) or VAPA + IRBIT (blue, green, dark yellow) and stimulated with F/I (filled columns). **N**–**P** The $Ca^{2+}$ dependence of the ANO1+VAPA (red, blue) treated with siAC6 (black, grean) and stimulated with F/I (blue, green) (**N**), the Vmax (**O**), and apparent Km for $Ca^{2+}$ (**P**). In (**A**–**C**, **M**) pipette solution, $Ca^{2+}$ was 0.3 μM. Results are from 6 (VAPA + siAC6) or 3 (all others) independent experiments. All results are shown as mean ± s.e.m.

ANO1, while VAPB had minimal effect, and VAPA + IRBIT increased the current much more than VAPB + IRBIT (Supplementary Fig. 3D, E and Fig. 3A). Finally, the knockdown of VAPB had a minimal effect on the basal current and did not affect the IRBIT-stimulated current (Supplementary Fig. 3F), while knockdown of VAPA eliminated the effect of IRBIT (Fig. 3A). Together, these results indicate that the regulation of ANO1 is primarily mediated by VAPA, and thus all subsequent studies are with VAPA.

Figure 3A (I/Vs in Supplementary Fig. 4A) shows that at a $Ca^{2+}$ concentration of 0.3 μM, F/I stimulation inhibited the current activated

by VAPA. Although IRBIT did not increase the current beyond that measured with VAPA, IRBIT fully reversed the inhibition of ANO1 current by forskolin (open red column in Fig. 3A). The effects of VAPA on ANO1 current required IRBIT, as shown by the elimination of VAPA effects by mutation of the VAPA binding motif of IRBIT (Fig. 3B) and by deletion of IRBIT (Fig. 3C). These findings suggest that IRBIT and VAPA synergize in the activation of ANO1 and that this synergy requires the cAMP pathway. TIRF measurement revealed that VAPA markedly increased ANO1 in the TIRF field, that was further increased by IRBIT (Supplementary Fig. 4B), which can account in part for the effects of

VAPA alone and together with IRBIT on ANO1 current. Surprisingly, stimulation with F/I that inhibited the VAPA-activated ANO1 current (Fig. 3A) increased, rather than decreased, the ANO1 TIRF signal. This unexpected effect was further examined by a biotinylation assay (Fig. 3D–F). Thus, again, there is a complete correlation between TIRF and biotinylation assays in reporting the surface expression of ANO1. These findings indicate that the inhibition of ANO1 current by F/I is underestimated. The increased ANO1 surface by VAPA and F/I stimulation involved enhanced interaction between ANO1 and IRBIT as demonstrated by the increased ANO1-IRBIT FRET (Supplementary Fig. 4C). Together, the findings in Fig. 3 suggest that at least two PKA phosphorylation sites are involved in the effects of VAPA and VAPA + IRBIT on ANO1 function, an activatory and inhibitory sites. Measuring the current of ANO1(S673A) indicated that S673 was not involved in the effect of VAPA alone, but S673 mediated the synergy between VAPA + IRBIT (Supplementary Fig. 4D). Further analysis to be described below when discussing the role of E-Syt2 identified S221 as a residue mediating inhibition of ANO1 by PKA. When expressed with VAPA, ANO1(S221A) current was not inhibited by F/I stimulation (Supplementary Fig. 4E).

The effect of VAPA on ANO1 activity suggested that ANO1 has a VAPA binding motif that was identified in the N terminus of ANO1 [236]DKD**SFFDS**KTR[247] with a score of 1.5[43], and the mutation S243E disrupts the motif by increasing the score to 5.5[29,41]. ANO1(S243E) showed reduced expression in the TIRF field and prevented the effects of VAPA, IRBIT, and F/I stimulation on ANO1 TIRF signal (Supplementary Fig. 4F), and their proximity analyzed by FRET microscopy (Supplementary Fig. 4G). Accordingly, ANO1(S243E) exhibited a lower basal current than ANO1 and did not respond to VAPA, IRBIT, or stimulation by F/I (Supplementary Fig. 4H). Measurement of the $Ca^{2+}$ dependence of ANO1(S243E) current (Fig. 3G–I, and Supplementary Fig. 5A–C) confirmed the reduction in current Vmax and a modest reduction in app-Km.

The effect of ANO1(S243E) on ANO1 activity prompted us to determine the effect of VAPA alone and with IRBIT on the activation of ANO1 by $Ca^{2+}$ (Fig. 3J–L and Supplementary Fig. 5D–F). VAPA alone reduced the ANO1 app-Km for $Ca^{2+}$ from 1.04 to 0.07 μM without affecting Vmax (red traces and columns). In the presence of VAPA, stimulation with F/I prominently increased the app-Km to 4.94 μM $Ca^{2+}$ and increased Vmax from 12.0 to 20.7 pA/pF. When IRBIT was included in the presence of VAPA, the app-Km was further reduced to 0.02 μM $Ca^{2+}$ with a slight increase in Vmax to 14.7 pA/pF. Finally, stimulating the cells expressing VAPA and IRBIT with F/I resulted in an app-Km of 0.09 μM $Ca^{2+}$ and increased Vmax to 24.1 pA/pF. Interestingly, while VAPA alone and VAPA + IRBIT increased the TIRF signal of ANO1 (Supplementary Fig. 4B), they did not increase the current Vmax until stimulation with F/I (Fig. 3K). We interpret this to suggest that VAPA likely caused the accumulation of ANO1 at the TIRF field and that insertion of this ANO1 fraction in the plasma membrane required stimulation with F/I. The biotinylation results in Fig. 3D–F support such a possibility.

Although they are analyzed in further detail below, for clarity and association, we describe the role of the adenylyl cyclases (ACs), the $Ca^{2+}$-activated AC8, and the $Ca^{2+}$-inhibited AC6, on the function of VAPA in Fig. 3M–P. The $Ca^{2+}$-regulated ACs in HEKT cells and their depletion by siRNA are shown in Supplementary Fig. 5G, H. Figure 3M shows that at 0.3 and 10 μM $Ca^{2+}$, the depletion of AC8 inhibited the activation of ANO1 by VAPA and by VAPA + IRBIT. FRET measurements showed that AC8 increased the ANO1-IRBIT interaction and maximized the effect of VAPA on their interaction (Supplementary Fig. 5I). However, depletion of AC8 did not alter the basal ANO1-IRBIT FRET but eliminated the effect of VAPA on the FRET (Supplementary Fig. 5J). These findings indicate that AC8 mediates a significant portion of the PKA-dependent regulation of ANO1 by VAPA. The results in Fig. 3N–P (and Supplementary Fig. 5K–M) indicate that AC6 mediates the

reduced $Ca^{2+}$ activation of ANO1 in the presence of VAPA and F/I stimulation. Consequently, depletion of AC6 reduced Vmax from 20.2 to 10.3 pA/pF and eliminated further effects of F/I, while reducing app-Km from 3.1 to 0.52 μM.

Together, the findings with VAPA indicate that VAPA regulates ANO1 by affecting its regulation by PKA which governs both its expression level in the PM and its activation by $Ca^{2+}$. Moreover, VAPA and IRBIT act synergistically in the regulation of ANO1 function.

### Regulation of ANO1 by IRBIT and PKA takes place at the ER/PM junction: Role of E-Syt1

The ER localized VAPA targets proteins to MCS but it is not a tether protein because it lacks PM interacting motif. MCS are established by tether proteins, and primary tethers at the ER/PM junctions are the Extended Synaptotagmins (E-Syts), E-Syt1, E-Syt2, and E-Syt3[44,45]. To evaluate the role of the E-Syts, we measured ANO1 current at 0.3 μM $Ca^{2+}$, which revealed that expression of E-Syt1 increased the current that was further enhanced by stimulation with F/I. Expression of E-Syt2 had no effect on the basal current, but F/I stimulation inhibited ANO1 current, while expression of E-Syt3 had no apparent effect on ANO1 current (Fig. 4A). Conversely, Fig. 4B shows that depletion of E-Syt1 had no effect on basal ANO1 current but eliminated the effect of IRBIT and F/I stimulation, while depletion of E-Syt2 significantly increased the basal current, had no effect on the increase by IRBIT, and prevented the increase caused by F/I stimulation. Consequently, we focused on the roles of E-Syt1 and E-Syt2 in regulating ANO1 function.

The opposite and intricate effects of the E-Syts required a detailed probing of each. Starting with E-Syt1, Fig. 4C shows that E-Syt1 and IRBIT similarly increased ANO1 current, and E-Syt1 slightly increased the current in the presence of IRBIT. Biotinylation (Fig. 4D) and TIRF assays (Supplementary Fig. 6A) showed that these effects are mainly attributed to increased surface expression of ANO1. Co-IP (Fig. 4D) and FRET measurements (Supplementary Fig. 6B, C) indicated that IRBIT and E-Syt1 mutually increase their interaction with ANO1, which was further increased by F/I stimulation and required the VAPA binding motif of ANO1. Although both IRBIT and E-Syt1 (Fig. 4A, H) increased ANO1 current, and the current was further increased by F/I stimulation when ANO1 was expressed with both IRBIT + E-Syt1 F/I stimulation inhibited the current, likely by recruitment of an AC that inhibits ANO1 (see below). Deletion of IRBIT eliminated the function of E-Syt1, which was restored by re-expression of IRBIT (Fig. 4E). The results in Fig. 1H and 4E showed an excellent agreement between the effect of IRBIT on the native and expressed ANO1. Hence, Fig. 1H showed that the native ANO1 activity ratio with and without IRBIT was 2.72, and Fig. 4E showed that for the expressed ANO1, the ratio was 2.45. IRBIT was required for the effect of both E-Syts and VAPA on ANO1 expression at the TIRF field and rescued by re-expression of IRBIT (Supplementary Fig. 6D, E). Thus, the effects of IRBIT and E-Syt1 (and of IRBIT and VAPA) were mutually dependent. On the other hand, while E-Syt1 had no effect on the ANO1 current in the presence of VAPA, VAPA was required for E-Syt1 function (Supplementary Fig. 6F, G). E-Syt1 increased the current of ANO1(S673A), but F/I-induced activation of the current was eliminated (Supplementary Fig. 6H), further demonstrating the crucial role of S673 in the activation of ANO1 by PKA.

AC8 was required for the F/I effect of E-Syt1 on ANO1 current, as shown by the elimination of this effect upon depletion of AC8 (Fig. 4F). Depletion of AC8 reduced the FRET between IRBIT and ANO1, as well as the increased FRET caused by E-Syt1 (Supplementary Fig. 6I). Figure 4G shows that depletion of AC6 prevented the inhibition of ANO1 current by F/I observed in cells expressing both E-Syt1 and IRBIT (Fig. 4C). AC6 also affected the interaction of ANO1 with IRBIT, with depletion of AC6 preventing the increased FRET caused by F/I stimulation and by E-Syt1 (Supplementary Fig. 7A). In a previous study, we reported that the ER-localized anoctamin 8 (ANO8) functions as a tether at the ER/PM junctions, similar to E-Syt1[46]. Therefore, we tested the effect of ANO8

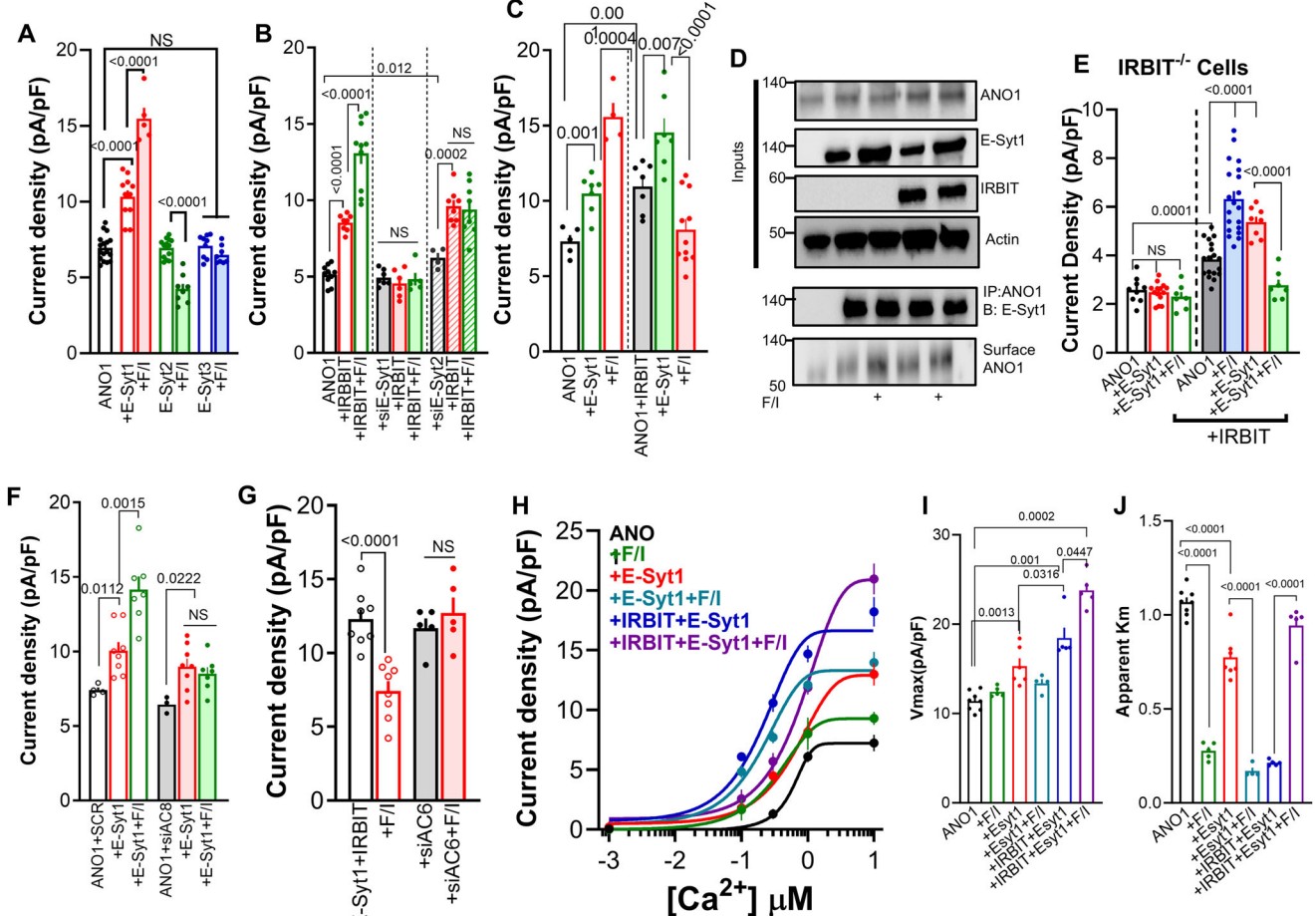

**Fig. 4 | Roles of E-Syt1 in regulation by the cAMP/PKA pathway of ANO1 activation by $Ca^{2+}$ and surface expression. A** Current was measured in cells expressing ANO1 (black) with E-Syt1 (red), E-Syt2 (green), or E-Syt3 (blue) and stimulated with F/I (filled columns). The results are from 4 independent experiments.
**B** Current was measured in cells treated with scrambled siRNA (open columns), siE-Syt1 (filled columns), or siE-Syt2 strips columns) and transfected with ANO1 (black) and IRBIT (red) and stimulated with F/I (green). The results are from 4 independent experiments. **C** Current was measured in cells expressing ANO1 (open columns) or ANO1 + IRBIT (close columns) and E-Syt1 (green) and treated with F/I (red). The results are from 3 independent experiments. **D** Effect of E-Syt1 and F/I on E-Syt1-ANO1 Co-IP and on surface ANO1 expression. This is one of 3 similar experiments. **E** Rescue by IRBIT of ANO1 current measured in $IRBIT^{-/-}$ cells, as indicated. The

results are from 5 independent experiments. **F** Current was measured in cells treated with scrambled siRNA (open columns) or siAC8 (filled columns) and expressing ANO1 and vector (black) or E-Syt1 (red) and treated with F/I (green). The results are from 3 independent experiments. **G** Current was measured with cells treated with scrambled (open columns) or siAC6 (filled columns) and expressing ANO1 and E-Syt1 + IRBIT (black) and treated with F/I (red). The results are from 4 independent experiments. **H–J** The $Ca^{2+}$ dependence of ANO1 in the presence of vector (black) stimulated with F/I (green), E-Syt1 (red) stimulated with F/I (turquoise), E-Syt1 + IRBIT (blue) stimulated with F/I (purple) (**H**), the Vmax (**I**) and apparent Km for $Ca^{2+}$ (**J**) were obtained from 5 independent experiments. Currents in (**A–D**, **F**, **G**) were measured with 0.3 μM $Ca^{2+}$. All results are shown as mean ± s.e.m.

on the ANO1 function. The results in Supplementary Fig. 7E–J show that ANO8 increased ANO1 current, which was further enhanced by F/I stimulation, and that a combination of IRBIT and ANO8 inhibited the current. ANO8 also increased ANO1 expression in the TIRF field, and the depletion of ANO8 eliminated the effects of IRBIT. Moreover, the ANO1(673 A) mutation and knockdown of AC8 prevented the effect of F/I stimulation. These findings indicate that E-Syt1 and ANO8 have a similar tether function in regulating ANO1 activity.

E-Syt1, both in the presence and absence of IRBIT, affected the activation of ANO1 by $Ca^{2+}$. The effects of E-Syt1 on the $Ca^{2+}$-dependent ANO1 current are shown in Fig. 4H–J (total current is in Supplementary Fig. 7B–D). In general, E-Syt1 amplified the effect of IRBIT. E-Syt1 alone increased the Vmax from 11.4 to 15.3 pA/pF, and in combination with IRBIT, it was further increased to 18.45 pA/pF. Subsequent F/I stimulation increased the Vmax to 23.78 pA/pF (Fig. 4I). As expected, F/I stimulation had a prominent effect on the app-Km for $Ca^{2+}$, reducing it from 1.07 to 0.27 μM. E-Syt1 alone reduced the app-Km to 0.77 μM, which was further reduced by F/I to 0.17 μM $Ca^{2+}$. Finally, IRBIT, in the

presence of E-Syt1, lowered the app-Km from 0.77 to 0.21 μM, that was increased by F/I stimulation to 0.94 μM $Ca^{2+}$. Evidently, multiple and different ACs are recruited by E-Syt1 and IRBIT individually and in combination to gate the activation of ANO1 by $Ca^{2+}$. This is addressed in detail further below.

### Regulation of ANO1 by IRBIT and PKA takes place at the ER/PM junction: Role of E-Syt2

Although E-Syt1 and E-Syt2 localize at the ER/PM junctions and heteromerize[44], they have opposite effects on ANO1 function (Fig. 4A). To understand the regulation of ANO1 by E-Syt2, we examine the complex formed between ANO1 and the proteins depicted on the left side of the model in Fig. 2M. When ANO1 current was measured at 0.3 and 10 μM $Ca^{2+}$ and in the presence of IRBIT, E-Syt2 inhibited the current (Fig. 5 I/V is in Supplementary Fig. 8A). Biotinylation (Fig. 5B) and TIRF measurements (Supplementary Fig. 8B) showed that E-Syt2 increased surface ANO1 and F/I stimulation reduced surface ANO1 levels. This suggests that E-Syt2 inhibits ANO1 current even without F/I

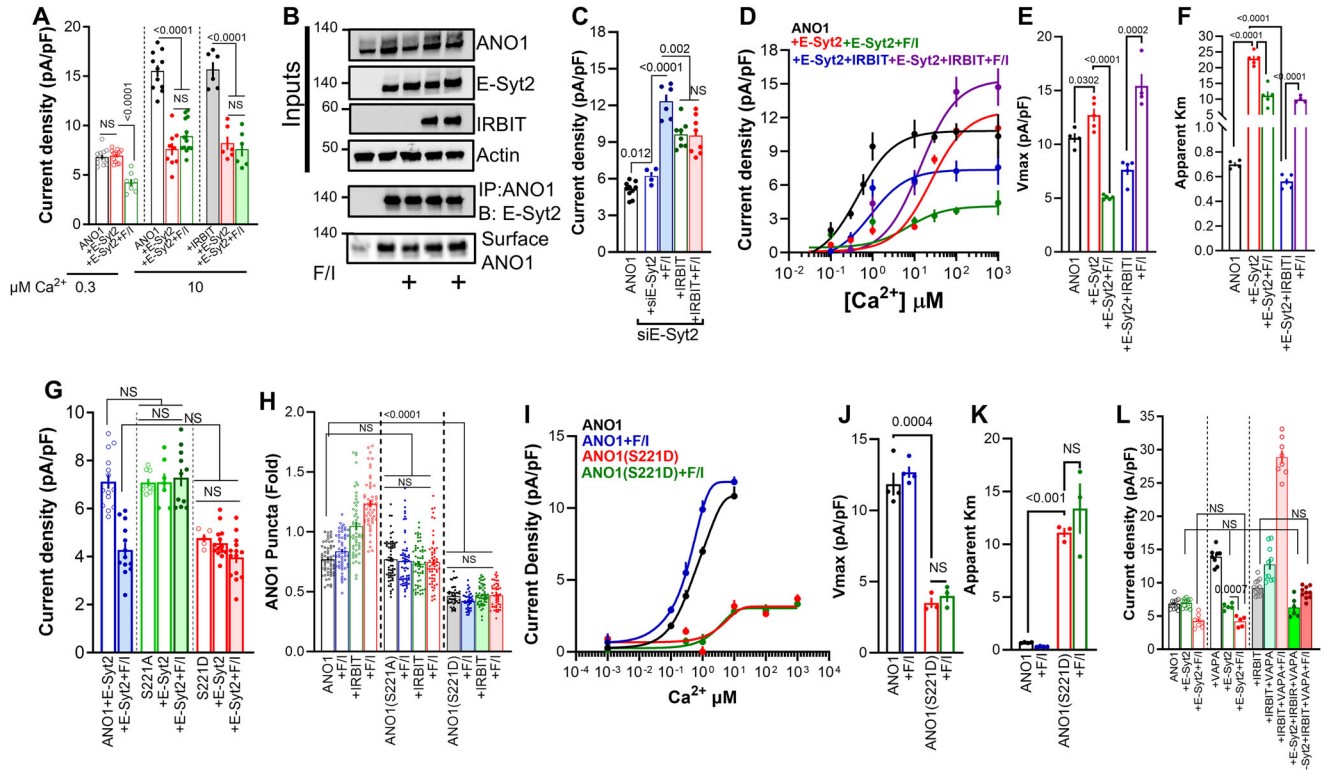

**Fig. 5 | Roles of E-Syt2 in regulation by the cAMP/PKA pathway of ANO1 activation by Ca²⁺ and surface expression. A** Current was measured in 4 independent experiments in the presence of 0.3 (open columns) or 10 μM pipette Ca²⁺ (closed symbols/open and filled columns) in cells expressing ANO1 (black) and E-Syt2 (red) treated with F/I (green). In all experiments with the filled columns, the cells were also transfected with IRBIT. **B** Effect of E-Syt2 and F/I on E-Syt2-ANO1 Co-IP and on surface ANO1 expression. This is one of 3 similar experiments. **C** Current was measured in 3 independent experiments in cells treated with scrambled siRNA (black) or siE-Syt2 (all others) and expressing ANO1 (black, blue), IRBIT (green, red), and treated with F/I (filled columns). **D–F** The Ca²⁺ dependence of ANO1 in 4 independent experiments in the presence of vector (black), and in 5 independent experiments with E-Syt2 (red) and stimulated with F/I (green), E-Syt2+IRBIT (blue) and stimulated with F/I (purple) (**D**), the Vmax (**E**) and apparent Km for Ca²⁺ (**F**). **G** Current was measured in 5 (or 4 with S221D) independent experiments with cells expressing ANO1 (blue), ANO1(S221A) (green) or ANO1(S221D) (red) and E-Syt2 and treated with F/I (filled columns). **H** TIRF was used in 6 independent experiments to measure ANO1 (open symbols and columns), ANO1(S221A) (close symbols/open columns), and ANO1(S221D) (filled columns) puncta in the presence of vector (black) and IRBIT (green) and stimulated with F/I (blue, red). **I–K** The Ca²⁺ dependence of ANO1 (black, blue, the same data as in Fig. 2A–C) and ANO1(S221D) (red, green) and stimulated with F/I (blue, green) (**I**), the Vmax (**J**) and apparent Km for Ca²⁺ (**K**) were obtained from 3 independent experiments. **L** ANO1 current was measured in 4 independent experiments (black) with E-Syt2 (green) and stimulated with F/I (red) in cells that, in addition, express VAPA (close symbols/open columns), and in the filled columns IRBIT (black), IRBIT + VAPA (light green) stimulated with F/I (red) and E-Syt2 + VAPA + IRBIT (dark green) stimulated with F/I (dark red). Unless otherwise indicated, currents were measured with 0.3 μM Ca²⁺. All results are shown as mean ± s.e.m.

stimulation, a finding supported by the increased ANO1 current in cells treated with siE-Syt2 (Fig. 5C). In addition, E-Syt2 increased the interaction between ANO1 and IRBIT (Supplementary Fig. 8C). In IRBIT⁻/⁻ cells, E-Syt2 slightly increased the current, and the inhibition by F/I stimulation was eliminated, which was fully rescued by re-expression of IRBIT (Supplementary Fig. 8D). Depletion of E-Syt2 increased the basal current, and F/I stimulation now markedly activated the current (Fig. 5C). The findings so far suggest that E-Syt2 recruits an inhibitory AC to ANO1, in a process that requires IRBIT.

The effect of E-Syt2 on Ca²⁺-dependent activation of ANO1 is shown in Fig. 5D–F (total current is in Supplementary Fig. 8E–G). In agreement with increased E-Syt2 surface expression, E-Syt2 increased ANO1 Vmax from 10.6 to 12.7 pA/pF while markedly increasing the app-Km from 0.7 to 22.8 μM Ca²⁺. F/I stimulation, which reduced surface expression, decreased Vmax to 5.1 pA/pF, and reduced app-Km from 22.8 to 11.1 μM Ca²⁺. In the presence of E-Syt2, IRBIT reduced Vmax from 12.7 to 7.6 pA/pF and lowered app-Km from 22.8 to 0.56 μM Ca²⁺. Finally, stimulation of cells expressing both E-Syt2 and IRBIT with F/I fully restored Vmax to 15.4 pA/pF but increased app-Km from 0.56 to 9.9 μM Ca²⁺. Although the fittings to obtain the app-Km values were not very tight, they are sufficient to conclude that the primary effect of

E-Syt2 is changing the ANO1 apparent affinity for Ca²⁺. In addition, it appears that the various combinations of E-Syt2 and IRBIT form complexes with distinct ACs to differentially determine the surface expression level of ANO1 to control ANO1 open probability and its activation by Ca²⁺.

To identify the residue phosphorylated by PKA that mediates inhibition of ANO1 by E-Syt2, we screened for mutants exhibiting minimal response to the increase in Ca²⁺ concentration from 0.3 to 1 μM, a response typically observed in cells expressing E-Syt2. Supplementary Fig. 8H shows that the S221A and T258A mutants exhibit such characteristics. ANO1(T258A) showed significantly increased basal current and current inhibition in response to the expression of IRBIT (Supplementary Fig. 8I), both of which differ from the effects observed with E-Syt2 (Fig. 5A, D). Therefore, we focused on the properties of ANO1 S221. Figure 5G shows that ANO1(S221A) did not affect the basal current, was not influenced by E-Syt2, and was not inhibited by F/I stimulation. On the other hand, the phosphomimetic ANO1(S221D) exhibited a current density similar to that observed with F/I-stimulated ANO1 + E-Syt2, and it was not affected by F/I stimulation. The effects of the S221 and S221D mutants on the ANO1 level in the TIRF field (Fig. 5H) suggest that their effect on basal current is likely due to their effects on the ANO1 surface level. To further illustrating

the role of S221, we tested the activation of ANO1(S221D) by $Ca^{2+}$, predicting that it would behave similarly to ANO1 in the presence of both E-Syt2 and F/I. Figure 5I–K shows that ANO1(S221D) current displayed a markedly reduced Vmax (3.5 pA/pF) and a high app-Km (11.1 μM $Ca^{2+}$), comparable to the values measured for E-Syt2+F/I (green in Fig. 5D–F) and it was not altered by F/I stimulation.

Notably, E-Syt2 function dominated over VAPA by inhibiting the increased current induced by VAPA, in the presence and absence of IRBIT, when measured at 0.3 (Fig. 5L) or 10 μM $Ca^{2+}$ (Supplementary Fig. 9A), and the ANO1(S243E) VAPA mutant (Supplementary Fig. 9B). Moreover, E-Syt2 dissociated the VAPA-IRBIT-ANO1 complex measured by FRET (Supplementary Fig. 9C, D). Together, these findings suggest that E-Syt2 exerts an effect opposite to that of E-Syt1, controlling ANO1 function by disrupting ANO1 interaction with VAPA and promoting the formation of an alternative complex with the ACs. These results in reduced ANO1 surface expression and a marked decrease in ANO1 affinity for $Ca^{2+}$.

## Regulation of ANO1 by IRBIT and PKA takes place at the STIM1 ER/PM junction

A key protein forming the ER/PM junctions is STIM1. STIM1 acts as a dynamic tether with ER transmembrane domain and a folded cytoplasmic domain that unfolds and dimerizes in response to $Ca^{2+}$ release from the ER, thereby forming new ER/PM junctions[47]. The main function of STIM1 at the junctions is to cluster and activate the Orai1 channels. However, its involvement in the regulation of the cAMP/PKA pathway has not been previously examined in any detail. We asked if STIM1 has a role in the assembly of the cAMP/PKA signaling pathway and the transduction of the cAMP/PKA signal, as illustrated in the model in Fig. 2M. To investigate this, we measured the ANO1 current in STIM1$^{-/-}$ cells. The deletion of STIM1 did not affect the basal current and the increase induced by IRBIT but eliminated the current increased by F/I stimulation (Fig. 6A, B). Moreover, STIM1 deletion eliminated all effects of E-Syt1 on ANO1 current: the increase in basal current, the augmentation of the IRBIT effect, and the inhibition caused by F/I stimulation in the presence of IRBIT and E-Syt1 (compare Figs. 4C and 6A). The impact of STIM1 deletion on the current correlated with its effect on the ANO1 levels at the TIRF field (Fig. 6C). By contrast, Fig. 6D shows that the deletion of STIM1 did not affect the response to VAPA alone but converted the VAPA + IRBIT-induced increase in current to a reduction, eliminating the marked synergistic increase observed with F/I stimulation in cells expressing VAPA + IRBIT (Compare Figs. 3A and 6D). However, the current changes did not correspond to the effect of VAPA + IRBIT on ANO1 levels at the TIRF field in STIM1$^{-/-}$ cells (Fig. 6E), suggesting that the current reduction induced by VAPA + IRBIT is underestimated and primarily due to inhibition of channel conductance and NPo. Finally, STIM1 deletion eliminated the F/I-mediated effects on ANO1-IRBIT interaction and the effects of E-Syt1 and E-Syt2 on this interaction but did not affect the interaction with VAPA (Fig. 6F).

The analysis of the $Ca^{2+}$ dependence of ANO1 activation in STIM1$^{-/-}$ cells (Fig. 6G–I and Supplementary Fig. 10A–C) shows that disrupting STIM1 junctions did not affect the IRBIT-mediated increase in Vmax or app-Km but eliminated the effect of F/I stimulation. Interestingly, STIM1 deletion uncovered an ANO1 stimulation function by E-Syt2 that was inhibited by F/I stimulation and had no effect on the current measured in the presence of IRBIT (compare Figs. 6J and 5D–F). Since IRBIT is required for the inhibitory function of E-Syt2 (Supplementary Fig. 8D), in Fig. 6K–M (and Supplementary Fig. 10D–F), we determined the role of STIM1 on the effect of E-Syt2 with and without IRBIT on the various modes of activation of ANO1 by $Ca^{2+}$. At cytoplasmic $Ca^{2+}$ below 0.3 μM, E-Syt2 increased ANO1 current that was nearly fully inhibited at cytoplasmic $Ca^{2+}$ above 1 μM, suggesting recruitment by E-Syt2 to ANO1 of a $Ca^{2+}$-inhibitable AC that reduced ANO1 activation by $Ca^{2+}$ (see below roles of AC6). Stimulating STIM1$^{-/-}$ cells expressing E-Syt2

with F/I eliminated this activity (green trace). IRBIT reversed the inhibition by E-Syt2, recovering both the Vmax and the reduced app-Km (Fig. 6L, M) that was not affected further by stimulation with F/I. Overall, the findings in Fig. 6 reveal a novel function of STIM1; facilitating cAMP signal transduction, a process strictly dependent on the localization and integrity of the STIM1 ER/PM junctions.

## The key ACs in the regulation of ANO1 by VAPA, IRBIT, and the E-Syts by PKA

The results so far point that complexes involving VAPA-IRBIT-E-Syts communicate with the cAMP/PKA pathway at the STIM1 junctions, determining ANO1 surface expression and $Ca^{2+}$affinity. To determine whether specific ACs mediate each function and their identity, we focused on the $Ca^{2+}$-dependent ACs, the $Ca^{2+}$-activated AC1, AC3, and AC8, along with the $Ca^{2+}$-inhibited AC5 and AC6[48]. All these ACs are expressed in the HEKT cells we used, and the efficiency of their knockdown by the most effective siRNA (of the three tested) are shown in Supplementary Fig. 5H. The overall results with the ACs were not straightforward but rather appeared to be context-dependent on the complement of the native ACs and with the native tethers present at the ER/PM junctions.

Figure 7A shows that each of the ACs affected the ANO1 function and its regulation by IRBIT. Hence, depletion of AC1 and AC6 had no effect on basal ANO1 current measured at 0.1 μM $Ca^{2+}$, whereas depletion of AC3 increased and depletion of AC8 decreased the basal current, suggesting that native AC3 is inhibitory and native AC8 is stimulatory ACs that constitutively inhibit and stimulate ANO1, respectively. Depletion of AC1 had no effect on the increase in ANO1 current by IRBIT but resulted in current inhibition by F/I, suggesting that native AC1 participates in the PKA-stimulated and IRBIT-dependent increase in the current. Depletion of AC3 prevented further effects of IRBIT and F/I stimulation on ANO1 current, suggesting that native AC3 is a global inhibitor of ANO1 by the cAMP/PKA pathway. Depletion of AC8 prevented the effects of IRBIT and F/I on ANO1 current, suggesting that native AC8 is a key activator of ANO1 current by the PKA pathway that is required even for the effect of IRBIT on Vmax of ANO1. Finally, depletion of AC6 had no effect on the current stimulated by IRBIT but increased the current stimulated by F/I, indicating a role of native AC6 in the inhibition of ANO1 current. It must be noted that the effects of depletion of each AC are measured in the presence and effects of the complement of the native ACs. Therefore, an important implication of the stimulatory and inhibitory effects of the ACs is that they must be selectively compartmentalized with ANO1. Although not studied as well as the AKAPs, compartmentalization of the AKAPs implies similar compartmentalization of the ACs[49]. The cAMP generated by the ACs allows the differential effects by the PKA catalytic subunits (PKAc). FRET measurements suggest that most of the activity is mediated by the membrane-associated PKAII since ANO1 association with PKA-RII is preferred on that with PKA-RI in the presence and absence of AKAP5 (Supplementary Fig. 10G).

We decided to primarily focus on AC8 and AC6 because they are $Ca^{2+}$-dependent, localize to the luminal membrane of epithelia[50,51], as the localization of ANO1[52], and are both regulated by $Ca^{2+}$ influx through STIM1-Orai1[53,54], similar to ANO1[55]. Figure 7B shows that AC8 alone increased the ANO1 current, and F/I had no further effect, likely because the expressed AC8 generated sufficient cAMP to maximize activation of ANO1 by PKA. AC8 also augmented the stimulatory effects of IRBIT and of IRBIT and F/I. The effects of AC8 were at least in part due to the increased ANO1 level at the TIRF field, that was further increased by E-Syt1 but suppressed by E-Syt2 (Supplementary Fig. 10H). At 0.3 μM $Ca^{2+}$, expression of AC6 had no effect on basal ANO1 activity but, like it was found with E-Syt2, F/I stimulation inhibited ANO1 current. Increasing $Ca^{2+}$ to 10 μM eliminated all effects of AC6 and F/I stimulation (Fig. 7C), consistent with the inhibition of AC6 by $Ca^{2+}$. Depletion of AC6 increased both the basal and F/I stimulated

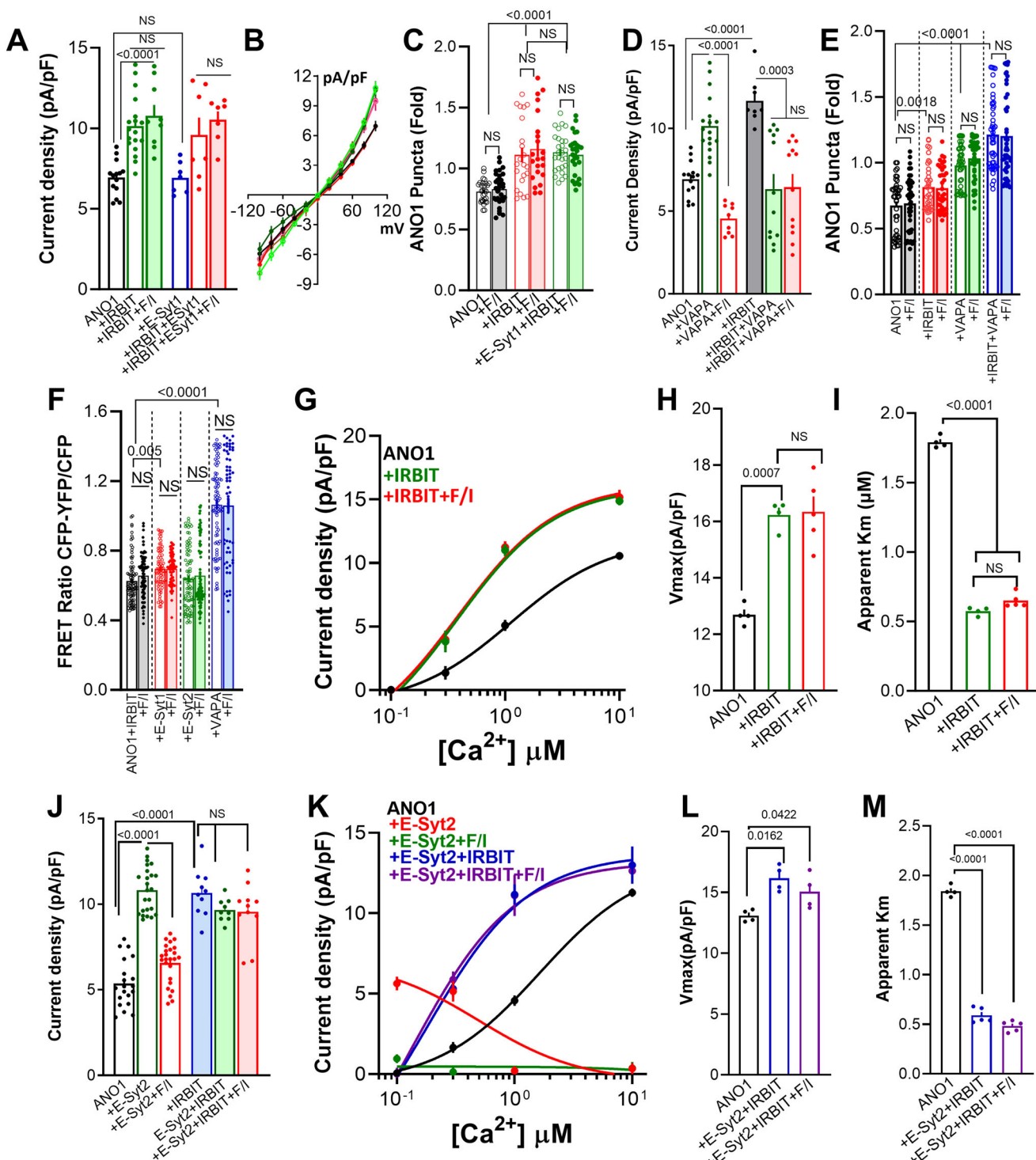

**All experiments in STIM 1 $^{-/-}$ cells**

ANO1 current, which was strongly reduced by concomitant depletion of AC6 and AC8 (Supplementary Fig. 10I). Expression of AC6 reduced the level of ANO1 at the TIRF field in cells expressing IRBIT, E-Syt1 or VAPA but not E-Syt2. The reduction in TIRF corelated with the reduction in surface ANO1 by expression of AC6 (Supplementary Fig. 10K). Accordingly, the depletion of AC6 increased the ANO1 level at the TIRF field, and the increase did not require the STIM1 junctions, but the deletion of STIM1 eliminated the effect of F/I stimulation (Supplementary Fig. 10L).

To identify the primary AC that mediates the inhibitory effect of E-Syt2, we searched for an AC the depletion of which prevents the inhibition of ANO1 current by F/I stimulation when expressed with E-Syt2. Measurements of ANO1 current at cytoplasmic $Ca^{2+}$ of 0.3 μM (Supplementary Fig. 11A) showed that depletion of AC1 reduced the basal current with E-Syt2, but the current was increased by F/I, while depletion of AC3 increased the current with E-Syt2 that was markedly inhibited by F/I stimulation. All the effects were reversed by co-depletion of AC1 and AC3. Current measurements at 0.3, 1 and 10 μM

**Fig. 6 | Roles of the STIM1 junctions in regulation of ANO1 by the cAMP/PKA pathway.** All experiments in Fig. 6 were conducted with STIM1$^{-/-}$ cells. **A, B** ANO1 current (black) was measured in 5 independent experiments in the presence of IRBIT (green), E-Syt1 (blue), IRBIT + E-Syt1 (red), and stimulation with F/I (filled columns). **C** Surface ANO1 (black) was measured in 4 independent experiments in the presence of IRBIT (red) or IRBIT + E-Syt1 (green) in cells stimulation with F/I (filled columns). **D** Current was measured in 4 independent experiments in cells expressing ANO1 (black) and VAPA (green) and stimulated with F/I (red) and IRBIT (filled columns). **E** Surface ANO1 (black) was measured in 5 independent experiments in the presence of IRBIT (red), VAPA (green), or IRBIT + VAPA (blue). **F** The ANO1-YFP and IRBIT-CFP FRET was measured in 7 independent experiments in cells

expressing ANO1 (black) and E-Syt1 (red), E-Syt2 (green) or VAPA (blue). **G–I** The Ca$^{2+}$ dependence of the ANO1 in the presence of vector (black), IRBIT (green), and stimulated with F/I (red) (**G**), the Vmax (**H**) and apparent Km for Ca$^{2+}$ (**I**) were obtained from 4 independent experiments. **J** ANO1 current was measured in 5 independent experiments with vector (black) and E-Syt2 (green) and treated with F/I (red) and in the presence of IRBIT (filled columns). **K–M** The Ca$^{2+}$ dependence of the ANO1 in the presence of vector (black), E-Syt2 (red) and stimulated with F/I (green), E-Syt2 + IRBIT (blue) and stimulated with F/I (purple) (**L**), the Vmax (**M**) and apparent Km for Ca$^{2+}$ (**N**) were obtained from 4 independent experiments. Unless otherwise indicated, currents were measured with 0.3 μM Ca$^{2+}$. All results are shown as mean ± s.e.m.

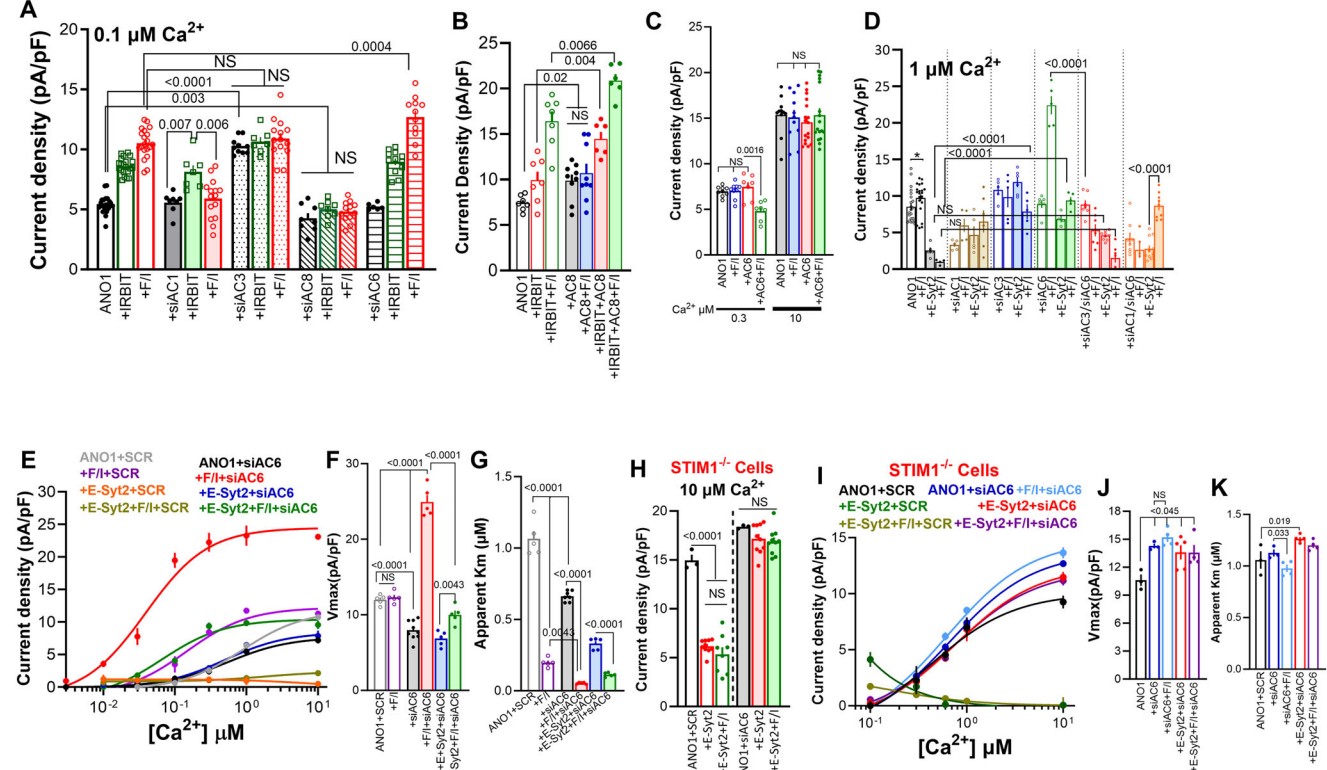

**Fig. 7 | Sensitization of ANO1 to Ca$^{2+}$ by AC8 and desensitization by AC6. A** Cells treated with scrambled siRNA (open columns), siAC1 (filled columns), siAC3 (dotted columns), siAC8 (diagonal strips), or siAC6 (horizontal strips), where used to measure ANO1 current (black) in the presence of IRBIT (green) and stimulated with F/I (red). Results are from 3 independent experiments. **B** Current was measured in 3 independent experiments in cells transfected with vector (open columns) or AC8 (filled columns) expressing ANO1 (black, blue), IRBIT (red), and stimulated with F/I (Green and Blue). **C** Current was measured in 4 independent experiments at 0.3 (open columns) or 10 μM Ca$^{2+}$ (close columns) in cells transfected with ANO1 and vector (black) or AC6 (red) and stimulated with F/I (blue, green). **D** The Ca$^{2+}$-dependent current was measured in 3 independent experiments at 1 μM Ca$^{2+}$ in cells that were treated with scrambled siRNA (black), siAC1 (brown), siAC3 (blue), siAC6 (green), siAC3 + siAC6 (red) or siAC1 + siAC6 (orange) and expressing ANO1 (open columns) and E-Syt2 (close columns). Cells were unstimulated (open symbols) or

were stimulated with F/I (close symbols). **E–G** The Ca$^{2+}$ dependence of the ANO1 was measured in 5 independent experiments in cells treated with scrambled (SCR) siRNA (gray) and stimulated with F/I (purple), E-Syt2 (orange), and stimulated with F/I (dark yellow). Cells were treated with siAC6 (black) and stimulated with F/I (red), E-Syt2 (blue), and stimulated with F/I (green) (**E**), the Vmax (**F**), and apparent Km for Ca$^{2+}$ (**G**). **H** Current was measured in 3 independent experiments at 10 μM Ca$^{2+}$ in STIM1$^{-/-}$ cells treated with scrambled (open columns) or siAC6 (filled columns) expressing ANO1 (black) and E-Syt2 (red) and stimulated with F/I (green). **I–K** The Ca$^{2+}$ dependence of the ANO1 was measured in 3 independent experiments in STIM1$^{-/-}$ cells treated with scrambled (SCR) siRNA, expressing E-Syt2 (green), and stimulated with F/I (dark yellow). Cells were treated with siAC6 (blue) and stimulated with F/I (light blue) expressing E-Syt2 (red) and stimulated with F/I (purple) (**I**), with the Vmax (**J**) and apparent Km for Ca$^{2+}$ (**K**). All results are shown as mean ± s.e.m.

Ca$^{2+}$ (Fig. 7D and Supplementary Fig. 11A–C) show that AC1, AC5, and AC6 may mediate the regulation of ANO1 by E-Syt2. However, AC1 is Ca$^{2+}$ activated, and the AC mediating the effects of E-Syt2 is Ca$^{2+}$-inhibitable (Figs. 5D and 6K). Moreover, at 1 μM Ca$^{2+}$ siAC1 nearly eliminated the stimulatory effect of F/I in siAC6 cells (Orange open column in Fig. 7D). To differentiate between the Ca$^{2+}$ inhibitable AC5 and AC6, we measured the effects of their depletion on ANO1 current at 10 μM Ca$^{2+}$ (Supplementary Fig. 11C). F/I stimulation of cells with depleted AC6 markedly increased the current measured at 1 and 10 μM Ca$^{2+}$

(green columns), which maybe mediated by a combination of AC1 and AC3 since it was eliminated by siAC3 in cells with depleted AC6 (red columns in Fig. 7D). However, depletion of AC5 inhibited all responses to F/I stimulation and to E-Syt2 (Supplementary Fig. 11A, C). Therefore, AC5 is not the main mediator of the inhibition caused by E-Syt2 while AC6 is, and thus, we analyzed the role of AC6 further.

Figure 7E–G (and Supplementary Fig. 12A–C) show the effect of depletion of AC6 on the Ca$^{2+}$-dependent ANO1 current. The controls with scrambled siRNA (SCR) and with E-Syt2 and F/I stimulation are

similar to those in Fig. 5D. Depletion of AC6 slightly reduced Vmax from 11.98 to 7.95 pA/pF and reduced app-Km from 1.06 to 0.66 μM $Ca^{2+}$. F/I stimulation of the AC6-depleted cells markedly increased Vmax from 7.95 to 24.9 pA/pF (300%), and reduced app-Km from 0.66 to 0.053 μM $Ca^{2+}$ (red in Fig. 7E–G). Clearly, AC6 is a potent inhibitor of ANO1 current. In AC6 depleted cells, E-Syt2 had no effect on the Vmax, increased the app-Km from 0.053 to 0.33 μM $Ca^{2+}$ (blue in Fig. 7E–G), and nearly eliminated the marked increase in Vmax by F/I stimulation while slightly reducing app-Km to 0.11 μM $Ca^{2+}$ (green in Fig. 7E–G). Thus, in the absence of AC6, E-Syt2 recruits another AC, perhaps AC1, AC3, or both (Fig. 7D, red and orange) to inhibit the activated ANO1 current. Thus, when recruited by E-Syt2 to ANO1, all the ACs inhibited the current, suggesting that the inhibitory function of the ACs is specified in part by E-Syt2, with AC6 being the primary inhibitory AC.

The profound effects of the depletion of AC6 on the ANO1 current were dependent on intact STIM1 junctions. In $STIM1^{-/-}$ cells, E-Syt2 inhibited the ANO1 current measured at 10 μM $Ca^{2+}$, and the inhibition was eliminated by depletion of AC6 (Fig. 7H). Measurement of ANO1 activation by $Ca^{2+}$ in $STIM1^{-/-}$ cells (Fig. 7I–K, and Supplementary Fig. 12D–F) showed profound inhibition of the effect of F/I stimulation in AC6 depleted cells and on the effects of E-Syt2. While in the presence of native STIM1 F/I stimulation of AC6 depleted cells increased Vmax to 24.9 pA/pF and reduced app-Km to 0.053 μM $Ca^{2+}$ (Fig. 7E–G), in $STIM1^{-/-}$ cells the Vmax (15.2 pA/pF) and the app-Km (0.98 μM $Ca^{2+}$) were not affected by depletion of AC6 (Fig. 7I–K). However, in $STIM1^{-/-}$ cells, depletion of AC6 eliminated the inhibition of the current by E-Syt2 and F/I stimulation (compare Fig. 7E–G, I–K).

## Key AKAPs in the regulation of ANO1 by PKA
The PKA signaling pathway is organized by the adapters A-kinase anchoring proteins (AKAPs)[56]. There are more than 30 AKAPs known, but we decided to focus on three plasma membrane-associated AKAPs AKAP3 (also known as AKAP110), AKAP5 (also known as AKAP79), and AKAP11 (also known as AKAP220). These AKAPs were selected since AKAP3 and AKAP11 have FFAT motifs recognized by VAPA[27,57], and AKAP5 is activated by $Ca^{2+}$ influx through Orai1, as is ANO1[54]. Moreover, these AKAPs are found in epithelia[58–61] and some of which respond to $Ca^{2+}$[56]. Supplementary Fig. 13A, B show the relative levels of AKAPs 3, 5, and 11 in HEKT cells and the efficient knockout of mRNA of these AKAPs by siRNA. Depletion of AKAP3 and AKAP5 increased ANO1 current while depletion of AKAP11 had no effect (Fig. 8A), which were in part due to the effects of the AKAPs on ANO1 at the TIRF field (Fig. 8B). Figure 8C shows that IRBIT did not activate ANO1 current in AKAP3 depleted cells, which was reduced to basal current by F/I stimulation. IRBIT inhibited the current in AKAP5-depleted cells that was not affected further by F/I stimulation, and depletion of AKAP11 largely eliminated the stimulatory effect of IRBIT but retained inhibition by F/I stimulation. Accordingly, the depletion of AKAP3 increased ANO1 at the TIRF field in the presence of IRBIT and the IRBIT/ANO1 FRET, while the depletion of AKAP5 and AKAP11 largely prevented the effect of F/I (Supplementary Fig. 13C, D). These findings suggest that AKAP5, but not AKAP3 and AKAP11, is required for activation of the current by IRBIT.

Furthermore, the AKAPs also altered the function of VAPA, E-Syt1 and E-Syt2. VAPA inhibited the increased ANO1 current in AKAP3-depleted cells, cause stimulation by F/I of the ANO1 current in AKAP5-depleted cells and increased the ANO1 current in AKAP11-depleted cells, which was then inhibited by F/I stimulation (Fig. 8D). The effects of VAPA on the current correlated with effects on ANO1 level at the TIRF field (Fig. 8E). The effects of E-Syt1 on the ANO1 current are shown in Fig. 8F and were a mixture of the effects of VAPA and IRBIT. E-Syt1 inhibited the current in AKAP3 (like VAPA) and in AKAP5 (like IRBIT) depleted cells and increased the current in AKAP11 depleted cells (like VAPA). Yet, the effects of E-Syt1 on the current of ANO1 paralleled those on the changes in ANO1 at the TIRF field and E-Syt1 interaction

with ANO1 (Supplementary Fig. 13E, F). Hence, in addition to coupling ANO1 with ACs and PKA, the AKAPs tested appear to control primarily the tether functions of IRBIT, VAPA, and E-Syt1 in controlling ANO1 at the TIRF field and, thus, likely surface expression.

Deviation between effects of the ANO1 current and surface expression were found with E-Syt2 (Fig. 8G, H and Supplementary Fig. 13G, H). E-Syt2 inhibited the ANO1 current and the effect of F/I on the current in AKAP3 and AKAP5-depleted cells and had no further effect of the current in AKAP11-depleted cells (Fig. 8G). Yet, E-Syt2 had no effect of ANO1 levels at the TIRF field in the AKAPs depleted cells and converted the inhibitory effect of F/I stimulation to stimulatory effects in the AKAP5 and AKAP11 cells with minimal effect of depletion the AKAPs on the ANO1-E-Syt2 interaction (Supplementary Fig. 13G, H). Thus, E-Syt2 acts on the ANO1 function, rather than the surface expression of ANO1 to modulate the current. This can be due to a reduction in channel open probability and/or in channel conductance. Since the depletion of AKAP11 eliminated the E-Syt2 function (Fig. 8G) we tested if the expression of AKAP11 reproduces the E-Syt2 function. Figure 8H shows that expression of AKAP11 inhibited the basal ANO1 current that was not affected further by F/I stimulation. In addition, the expression of AKAP11 in cells expressing ANO1 and E-Syt2 reduced the ANO1 current similar to F/I stimulation. Together, the effects of the depletion and expression of AKAP11 indicate that AKAP11 mediates the modulatory function of E-Syt2 on ANO1 current.

## The role of lipid transfer by the E-Syts SMP domains
In addition to functioning as tethers, the E-Syts have lipid transfer SMP domains that transfer lipids between membranes[45,62]. Previous measurements using isolated SMP domains or full-length E-Syts reconstituted into phospholipid vesicles reported broad lipid and glycerophospholipid transfer with no obvious substrate specificity[63–65]. The SMP domains dimerize to form a hydrophobic tunnel for lipids transfer, with mutations in critical valine and leucine/isoleucine residues within the tunnel inhibiting lipid transfer[66]. We prepared the lipid transfer SMP mutants E-Syt1$^{V169W/L308W}$ and E-Syt2$^{V197W/I337W}$ to examine the role of lipid transfer by the E-Syts in their function. Given that PI(4,5)$P_2$ depletion eliminated the effect of IRBIT and F/I on ANO1 function (Fig. 2K) and that the coupling of PM PI(4,5)$P_2$ and PI(4)P levels to PtdSer transfer regulates STIM1 function[37,67], we analyzed the effect of the E-Syts and their SMP domain mutants on their and on ANO1 junctional localization and on lipids composition at ER/PM junctions using biosensors[68] (Supplementary Fig. 14). Inhibition of lipid transfer by the SMP domains reduced junctional expression of the E-Syts by about 50% but with no effect on that of ANO1 (Supplementary Fig. 14A, B). E-Syt1 increased the level of PI(4)P and PI(4,5)$P_2$ at the ER/PM junctions with minimal effect on the level of PtdSer, which were prevented by E-Syt1$^{V169W/L308W}$ that also reduced junctional PtdSer.

E-Syt2 decreased PI(4)P, PI(4,5)$P_2$, and PtdSer lipids levels at the ER/PM junctions. Although its junctional level was significantly reduced, surprisingly, the E-Syt2$^{V197W/I337W}$ mutant markedly increased the levels of all three lipids (Supplementary Fig. 14C–E). Since the E-Syts heterodimerize[69] we reasoned that E-Syt2$^{V197W/I337W}$ may function as the dominant negative of E-Syt2 and reverse an inhibitory effect of E-Syt2 on the function of E-Syt1. Indeed, depleting E-Syt1 prevented the increase in PM lipids caused by E-Syt2$^{V197W/I337W}$ (Supplementary Fig. 14F–H). Moreover, depletion of E-Syt2 increased the level of PI(4)P and PtdSer, similar to Syt2$^{V197W/I337W}$, which was inhibited by co-depletion of E-Syt1 (Supplementary Fig. 14I, J). Hence, E-Syt1 and E-Syt2 act on the same pool of PI(4)P, PI(4,5)$P_2$, and PtdSer to reciprocally determine their steady-state level at the ER/PM junctions.

Next, we measured the effect of the SMP mutants on the $Ca^{2+}$ dependence of ANO1 current. Figure 9A–C shows that E-Syt1$^{V169W/L308W}$ not only prevented the effect of E-Syt1 on the current but markedly inhibited activation of ANO1 by $Ca^+$ and regulation by F/I, as was found with cells expressing E-Syt2 and stimulated with F/I (green in Fig. 5D–F)

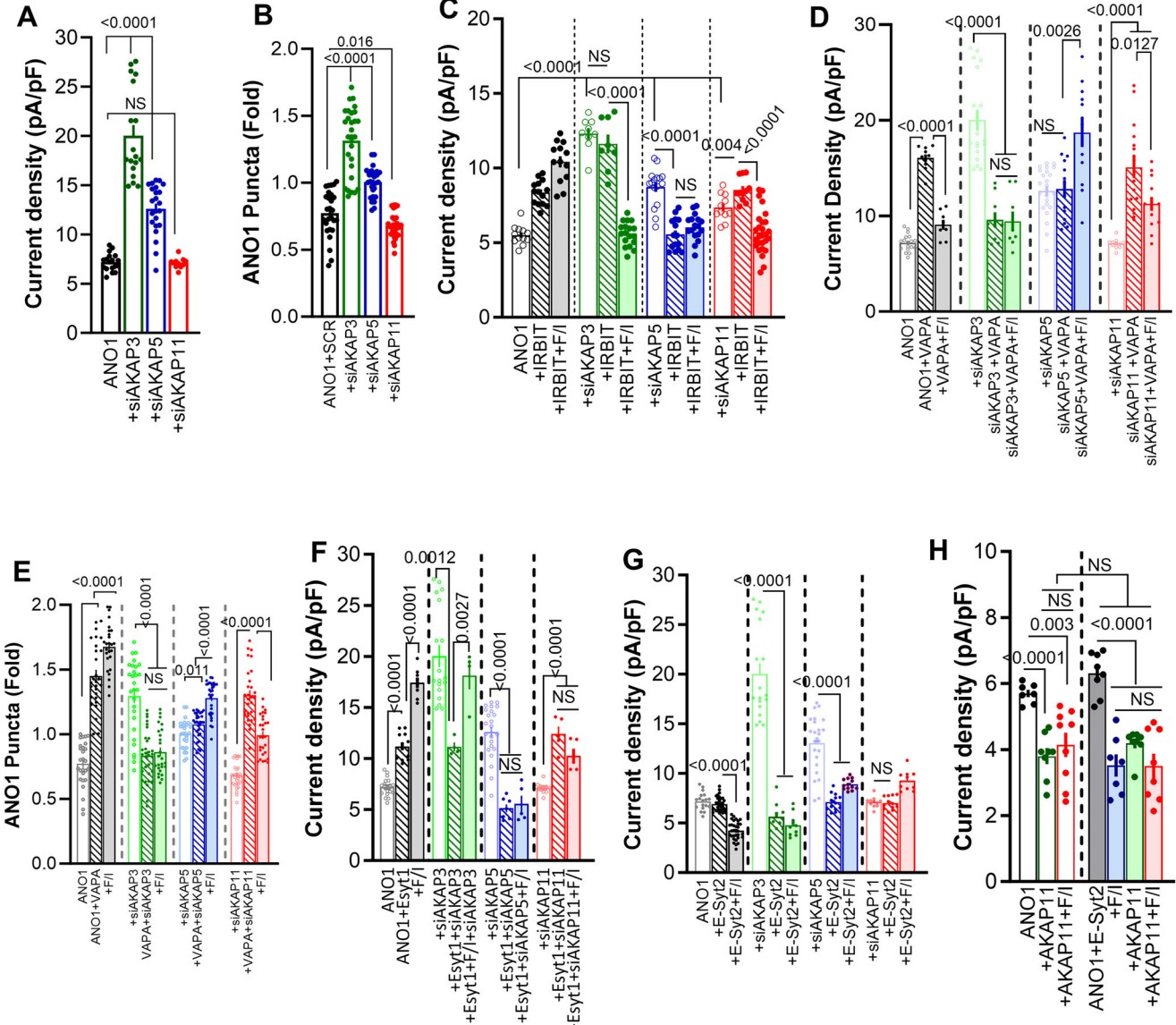

**Fig. 8 | The AKAPs mediating regulation of ANO1 by the cAMP/PKA pathway.**
**A–G** Cells were treated with scrambled siRNA (Black), siAKAP3 (green), siAKAP5 (blue) or siAKAP11 (red) and were used to measure (**A**), ANO1 current in (n = 4) (number of independent experiments in parenthesis). **B** ANO1 surface expression (n = 3). **C** ANO1 current (n = 6) with IRBIT and stimulated with F/I. **D** ANO1 current (n = 5) with VAPA (striped columns) and stimulated with F/I (filled columns). The currents in the absence of VAPA are the same as in (**A**). **E** Surface ANO1 (n = 3) when expressed with VAPA (striped columns) with cells stimulated with F/I (filled columns). **F** ANO1 current (n = 4) with E-Syt1 (striped columns) and stimulated with F/I (filled columns). **G** ANO1 current (n = 5) with E-Syt2 (striped columns) and stimulated with F/I (filled columns). **H** Current was measured (n = 4) in cells expressing ANO1 and vector (open columns) or ANO1 + E-Syt2 (close columns) (black) and together with AKAP11 (green) and stimulated with F/I (red and blue). Unless otherwise indicated, all currents were measured with 0.3 μM Ca²⁺. All results are shown as mean ± s.e.m.

and with ANO1(S221D) (red, green in Fig. 5I–K). On the other hand, E-Syt2^V197W/I337W prominently increased the App-Km of ANO1 for Ca²⁺ without affecting Vmax that was not affected by F/I stimulation (Fig. 9D–F). Moreover, the depletion of E-Syt1 that eliminated the effect of E-Syt2^V197W/I337W on the lipids eliminated the current activated by E-Syt2^V197W/I337W (Fig. 9G). These findings suggest that the reciprocal changes in PM lipids by the E-Syts have a major and perhaps a primary role in the activation of ANO1 by Ca²⁺ and regulation by the cAMP/PKA pathway.

As tethers, the ER-anchored E-Syts have to interact with the PM to form ER/PM junctions. Previous studies have shown that the fifth and third C2 domains of E-Syt1 and E-Syt2, respectively, interact with PM PI(4,5)P₂[44,70]. Accordingly, Fig. 9H shows that the deletion of the E-Syt1 C2E domain slightly reduced the basal current and eliminated the

effect of F/I. In addition, mutating the positively charged ^810RRK^812 PI(4,5)P₂-binding residues in E-Syt2 C2C domains (E-Syt2(KKR/AAA) increased the ANO1 basal current and prevented inhibition by F/I.

## Native ANO1 at the ER/PM junctions and regulation by the PKA pathway

To extend the physiological significance of the findings in Fig. 1 with the IRBIT^−/− mice and the molecular finings in HEKT cells to in vivo, we assay the effect of changing expression of the key proteins in Fig. 2M on the native ANO1 function in the immortalized human salivary gland acinar cell line NS-SV-AC[71,72]. We used immunoblotting to verify the expression of ANO1, IRBIT, VAPA, E-Syt1, E-Syt2, AC8 and AC6 in NS-SV-AC cells (Supplementary Fig. 15). The dominant Cl⁻ current in salivary gland acinar cells is mediated by ANO1[32] and, as expected, NS-SV-AC showed a

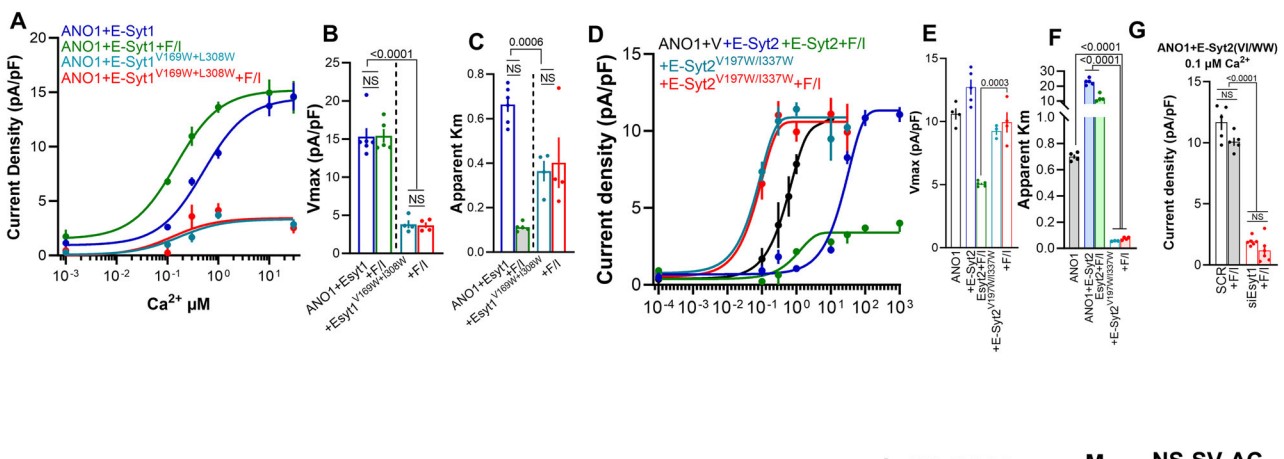

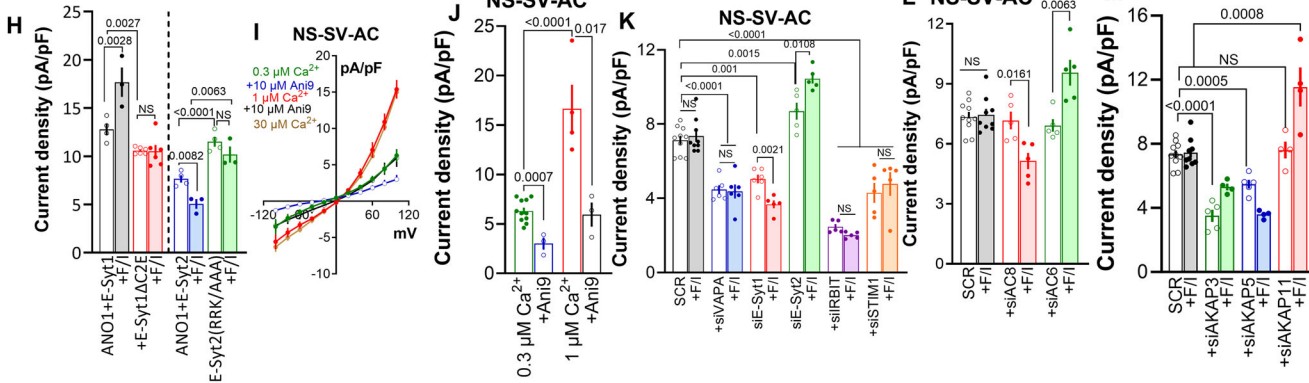

**Fig. 9 | The role of lipid transfer by the E-Syts SMP domains in activation of ANO1 and the role of the ER/PM junction tethers, ACs and AKAPs in the function of native ANO1 activity.** The $Ca^{2+}$ dependence of ANO1 current was measured in 4 independent experiments (**A**–**F**) in cells expressing: (**A**–**C**), E-Syt1 (blue), the SMP mutant E-Syt1$^{V169W/L308W}$ (turquoise) and stimulated with F/I (green, red); (**D**–**F**) vector (black), E-Syt2 (blue), the SMP mutant E-Syt2$^{V197W/I337W}$ (turquoise) and stimulated with F/I (green, red). Vmax (**B**, **E**) and apparent Km for $Ca^{2+}$ (**C**, **F**) are shown. The currents with E-Syt1 are from Fig. 4H–J, and with vector and E-Syt2 are from Fig. 5D–F. **G** ANO1 current was measured in 3 independent experiments in a pipette solution containing 0.1 μM $Ca^{2+}$ and expressing ANO1 + E-Syt2$^{V197W/I337W}$ and treated with SCR or E-Syt1 siRNA. **H** ANO1 current was measured in 3 independent experiments in a pipette solution containing 0.3 μM $Ca^{2+}$ and expressing the C2 domain mutants of E-Syt1 and E-Syt2, as indicated. (**I**–**M**), $Cl^-$ current was measured in the immortalized human salivary glands acinar cell line NS-SV-AC. **I**–**J** The native $Cl^-$ current was measured in 5 (green) or 3 (all others) independent experiments at 0.3 (green, blue), 1 (red, black), and 30 μM $Ca^{2+}$ (brown) and in the presence of the ANO1 inhibitor 10 μM Ani9 (black, blue). **K**–**M** ANO1 $Cl^-$ current was measured in 3 independent experiments at 0.3 μM $Ca^{2+}$ in NS-SV-AC cells treated with (**K**), scrambled (SCR) siRNA (black), or siRNAs targeting VAPA (blue), E-Syt1 (red), E-Syt2 (green), IRBIT (purple) or STIM1 (orange); (**L**), AC8 (red) or AC6 (green); (**M**), AKAP3 (green), AKAP5 (blue), AKAP11 (red) and stimulated with F/I (filled columns). All results are shown as mean ± s.e.m.

$Cl^-$ current that was activated by $Ca^{2+}$ and inhibited by the ANO1 inhibitor Ani9[34] (Fig. 9I–J). When measured across a range of $Ca^{2+}$ concentrations (0.3–30 μM), the native ANO1 current in NS-SV-AC cells showed characteristic voltage- and time-dependent behavior (Supplementary Fig. 14K) with outward rectification (Fig. 9I, J). Knockdown of the tethers VAPA and E-Syt1 significantly reduced native ANO1 current, with further inhibition observed upon F/I stimulation in E-Syt1 depleted cells (Fig. 9K). On the other hand, the depletion of E-Syt2 increased the basal current of the native ANO1, which was further increased by F/I stimulation (Fig. 9K). Depletion of STIM1 had the same effect as depletion of VAPA, while depletion of IRBIT almost eliminated the current (Fig. 9K). Moreover, the depletion of AC8 exposed the inhibitory effect of F/I stimulation, and the depletion of AC6 increased the basal and F/I-stimulated native ANO1 current (Fig. 9L). Finally, the depletion of AKAP3 and AKAP5 reduced, and the depletion of AKAP11 increased the native ANO1 current (Fig. 9M). These findings underscore the critical role of the PKA pathway-mediated regulation of ANO1 through complexes formed by VAPA, E-Syt1, and E-Syt2 at the STIM1 junction.

## Discussion

ANO1 has multiple physiological roles and affects the function of almost every cell[1–3,73]. These roles are attributed to ANO1 function as

the primary $Ca^{2+}$-activated $Cl^-$ channel. Yet, many of these functions are synergically regulated by $Ca^{2+}$ and the cAMP signaling pathways[18,22,24] that are not attributed to the effects of the cAMP pathways on ANO1 activity. Here, we discovered and characterized a fundamental molecular mechanism for dynamic regulation of ANO1 surface expression, current, and gating by $Ca^{2+}$ that is mediated by the multiple components of the cAMP/PKA pathway, which take place at the STIM1 ER/PM junctions and is dependent on junctional phospholipids composition.

This study was initiated by an in vivo observation in which deletion of IRBIT inhibited rather than the expected stimulation of ANO1 function and fluid secretion by salivary glands (Fig. 1). IRBIT was discovered as an inhibitor of the $IP_3Rs$[25] that competes with $IP_3$ for binding to the $IP_3Rs$ and inhibits $Ca^{2+}$ release from the ER[26]. Indeed, deletion of IRBIT in mice increased the receptor-evoked $Ca^{2+}$ signal in parotid acini. Parotid acini were selected for the studies since ANO1 is essential for fluid secretion by the salivary glands[5], and salivary fluid secretion requires both $Ca^{2+}$ and cAMP signaling provided by the parasympathetic and sympathetic inputs[9]. Although previous studies by our group showed that IRBIT activates ion transporters[17,74–77] and is involved in synergy between the cAMP and $Ca^{2+}$ signaling pathway in the ducts[17], it was expected that the increase in $Ca^{2+}$ signaling in the IRBIT$^{-/-}$ acini should increase ANO1 function in acinar cells since $Ca^{2+}$

release by the IP$_3$Rs and the activation of Ca$^{2+}$ influx were shown to couple to and activate ANO1. Particularly puzzling was the observation that deletion of IRBIT inhibited ANO1 function even at very high receptor stimulation when both Ca$^{2+}$ release and Ca$^{2+}$ influx were normal (Fig. 1C). This observation strongly indicated that IRBIT has functions beyond controlling Ca$^{2+}$ release by the IP$_3$Rs. The results provided in the present study indicate that IRBIT functions as a regulator of the ER/PM junctions that mediates multiple modes of Ca$^{2+}$ gating of ANO1 by the cAMP/PKA pathway.

Analysis of ANO1 and IRBIT motifs capable of mediating their interaction revealed that both have VAP-interacting FFAT motifs, and their interaction with VAPA is essential for the function of ANO1 and its communication with the cAMP/PKA pathway. Interestingly, VAPA was the primary regulator of ANO1 function while VAPB had a marginal role (Supplementary Fig. 3). VAPA and VAPB are paralogues and have 63% overall identity and 83% similarity, with their MSP domains that bind the FFAT motif have 82% identity and thus their functions are considered to overlap[78]. However, some specificity between the VAPs is likely since their interactome is not identical[42]. The effect of VAPA and not VAPB in the regulation of ANO1 is a clear example of the specific effect of the VAPs. VAPA regulates the expression of ANO1 at the plasma membrane (PM), its interaction with IRBIT, activation by Ca$^{2+}$ and communication with the cAMP/PKA pathways. The cooperation between IRBIT and VAPA is most noticeable when the cAMP/PKA system is engaged. In the absence of IRBIT, VAPA modestly increased Vmax and the apparent affinity for Ca$^{2+}$. F/I stimulation doubled the Vmax and markedly increased the affinity for activation of ANO1 by Ca$^{2+}$ (Fig. 3J–L). The effects of VAPA and IRBIT require AC8, and since AC8 (and the other ACs) does not have an obvious FFAT motif, IRBIT likely exerts its effects by promoting access of ANO1 to AC8 using other IRBIT domains.

It is noted that all components of the complexes formed by ANO1 are interdependent and are all required to form the complexes. Thus, disruption of the ANO1 FFAT motif, depletion of VAPA, deletion of IRBIT, depletion of E-Syt1, and even depletion of AC8 and of AKAP3 and AKAP5 affected the response to all other proteins. Conversely, increased expression of individual proteins increased the effect of most other proteins in the complexes. The most likely core proteins are VAPA and IRBIT. VAPA can oligomerize[42] to form a mesh that directly interacts with ANO1 and IRBIT FFAT motifs, which is then guided to the ER/PM junctions by E-Syt1. E-Syt1 does not possess a known FFAT motif but forms MCS at PI(4,5)P$_2$-rich domains in response to high cytoplasmic Ca$^{2+}$[44,64,79]. We name the VAPA-formed ANO1 core complex VANZ for the VAPA-Associated Nexus Zone. The classical ACs-AKAPs-PKA complexes appear to be localized within the STIM1-formed ER/PM junctions. Moreover, STIM1 is required for the communication between the ACs-AKAPs-PKA complexes and the VANZ/E-Syt1 complex to phosphorylate S673 and sensitize ANO1 to Ca$^{2+}$, as illustrated in Supplementary Fig. 16. The interaction of ANO1 with VAPA is antagonized by E-Syt2 that dissociates the ANO1-VANZ-IRBIT complex (Fig. 5). Like E-Syt1, E-Syt2 lacks an FFAT motif and is constitutively localized at the ER/PM junctions[44], where it facilitates the interaction of ANO1 with ACs/AKAPs/PKA complexes to phosphorylate S221, thereby desensitizing ANO1 to activation by Ca$^{2+}$ a process that requires IRBIT (Supplementary Fig. 16).

We encountered several difficulties in the definitive identification of the key ACs and AKAPs mediating ANO1 sensitization and desensitization. The first is that the same ACs could function as inhibitory and activatory, depending on the complement of the remaining ACs. For example, depletion of AC3 in wild-type cells resulted in activation of ANO1 that was no longer activated or inhibited by IRBIT or F/I stimulation, suggesting that activation of PKA by AC3 results in phosphorylation that inactivates ANO1 (Fig. 7A). By contrast, depletion of AC3 in AC6-deficient cells markedly inhibited ANO1 activity stimulated by F/I, indicating that when ANO1 is not phosphorylated by AC6/PKA AC3 is

required for the activation of ANO1 by phosphorylating the same or a different residue(s) compared to AC6 (Fig. 7D). In the case of the AKAPs, the issues are the large number of AKAPs (more than 30) and many of the AKAPs are expressed in multiple cellular sites[56]. Yet, extensive analysis of the Ca$^{2+}$ regulated ACs revealed two definitive; the Ca$^{2+}$-activated AC8 was required for all forms of sensitization of ANO1 to Ca$^{2+}$ and the Ca$^{2+}$-inhibited AC6 potently desensitized ANO1 to activation by Ca$^{2+}$ (Figs. 3–7) and both ACs and ANO1 respond to Ca$^{2+}$ influx by Orai1. With the AKAPs, we selected AKAP3 and AKAP11 since they possess FFAT sites[27,57] and AKAP5 since it is expressed at the STIM1 ER/PM junction[54]. AKAP3 was required for the activation of ANO1 by VAPA, and AKAP5 was required for activation of ANO1 by VAPA and E-Syt1, while AKAP11 mediated all actions of E-Syt2 (Fig. 8). Therefore, we propose that the likely (but not exclusive) complexes are VAPA-IRBIT-E-Syt1-AC8-AKAP5-PKA that phosphorylates S673 to sensitize and IRBIT-E-Syt2-AC6-AKAP11-PKA that phosphorylates S221 to desensitize ANO1 to Ca$^{2+}$ (Supplementary Fig. 16). Clearly, other cAMP/PKA pathway complexes are possible, increasing the diversity and cell-specific regulation of ANO1 depending on the complement of ACs and AKAPs expressed in each cell type.

In addition to regulating the activation of ANO1 by Ca$^{2+}$, VAPA, IRBIT, E-Syt1, E-Syt2, and the cAMP/PKA pathway regulate the PM level of ANO1, which matched the level of ANO1 at the TIRF field. VAPA, IRBIT, and E-Syt1 increase ANO1 surface expression independently of S673 phosphorylation (Figs. 2–4), while E-Syt2 reduces surface ANO1 through phosphorylation of S221. Phosphorylation of other serine/threonine residues by PKA may also regulate ANO1 surface expression and activity, as suggested by the inhibition of the IRBIT-mediated current increase upon mutation of several serine residues other than S673 and S221 (Supplementary Figs. 2C and 8H). We note that S673 is located close to the ANO1 Ca$^{2+}$ binding site[16]. S673 was identified as a residue phosphorylated by CaMKII to regulate the inhibition of ANO1 by ATP and PI(4)P$_2$[16]. However, clearly, PKA also phosphorylates this residue since it is required for F/I-stimulated increased ANO1 current, and the specific PKA inhibitor H89 eliminates the F/I-stimulated increase in ANO1 current (Fig. 2F). It is possible that the same residue can be phosphorylated by two kinases depending on cell type and cellular localization of ANO1, PKA and CaMKII and physiological needs. Significantly, we report here that PKA-mediated phosphorylation of S673 sensitizes ANO1 to Ca$^{2+}$ by increasing the affinity of ANO1 for Ca$^{2+}$. In addition, the cAMP/PKA-mediated increase in ANO1 at the TIRF field required phosphorylation of S673.

The E-Syts function as tethers but also as lipid transfer proteins by their SMP domains. The importance of the tether function was evident from the inhibition (but not reversal) of E-Syt1 and E-Syt2 function by inhibition of their C2 domains to bind to PM PI(4,5)P$_2$. Measurements of lipid binding and lipid transfer with reconstituted SMP domains of various LTP and in intact cells, including with E-Syt1 and E-Syt2, have reported their ability to bind and transfer multiple glycerophospholipids with limited substrate selectivity[80]. We focused on PI(4)P, PI(4,5)P$_2$, and PtdSer because of their important roles in STIM1 function at the ER/PM junctions[37,81]. Although we did not directly measure lipid transfer, we found that E-Syt1 and E-Syt2 have reciprocal effects on the steady-state level of these lipids at the junctions, with E-Syt1 increasing and E-Syt2 reducing their levels. The regulation of these lipids was concurrent, with E-Syt2 exerting dominance over E-Syt1, possibly because E-Syt2 is constitutively localized at the ER/PM junctions. These effects required functional SMP domains of both E-Syts (Supplementary Fig. 14). The most notable finding was that lipid transfer by the SMP domains determined the mode of ANO1 regulation by the E-Syts, as mutations in the SMP domain reversed their effects (Fig. 9). These findings have several implications: First, lipid transfer by E-Syts plays a more important role than tethering in determining their physiological function. Second, ANO1 senses the lipid environment and composition within the junctions that is at least partially regulated

by the E-Syts. Third, communication of the cAMP/PKA pathway with ANO1 and potentially other targets is regulated by junctional lipids.

The physiological implications of the present findings were demonstrated in an IRBIT$^{-/-}$ mouse model and in an immortalized human salivary gland acinar cell line (Figs. 1 and 9). These findings reinforce the important role of cAMP signaling in epithelial fluid secretion by salivary glands, exocrine pancreas, and other secretory epithelia where ANO1 activation by the Ca$^{2+}$ signaling pathway drives the secretion[9,82,83]. The model illustrated in supplementary Fig. 16 highlights the role of the cAMP pathway in tuning the function of ANO1 at the resting desensitized and activated sensitized states. E-Syt2, which is constitutively localized at ER/PM junctions, regulates the levels of PI(4)P, PI(4,5)P$_2$, and PtdSer. Regulation of these lipids by E-Syt2 is necessary for assembling the IRBIT/AC6/AKAP11/PKA complex. Moreover, E-Syt2 restricts the formation of the ANO1-VANZ-E-Syt1 complex to allow PKA to phosphorylate ANO1 serine 221. This phosphorylation reduces ANO1 surface expression and confers an ANO1 conformation with a low affinity for Ca$^{2+}$, maintaining ANO1 in a desensitized state. Cell stimulation that releases Ca$^{2+}$ from the ER results in STIM1 translocation along with E-Syt1 to ER/PM junctions. The STIM1 and E-Syt1 translocation increases junctional PI(4)P and PI(4,5)P$_2$ levels, facilitating the formation of the ANO1-VANZ-E-Syt1 complex and its association with the AC8-AKAP5-PKA complex, leading to the phosphorylation of ANO1 Serine 673. As a result, the reserve ANO1 channels close to the PM are inserted into the PM, and ANO1 adopts a sensitized conformation with high Ca$^{2+}$ affinity, becoming fully activated at 1 μM Ca$^{2+}$. Structural studies of ANO1 have revealed conformations with distinct Ca$^{2+}$ affinity[84,85]. In the low Ca$^{2+}$ affinity state, α6 helix is loose, and the channel is in the close state even if the Ca$^{2+}$ pocket is bound with 1 Ca$^{2+}$ ion. The binding of 1 Ca$^{2+}$ ion causes rearrangement of TM6 from an α to π conformation, resulting in increased affinity of the Ca$^{2+}$ binding site, binding of a second Ca$^{2+}$ ion, and perhaps enlargement of the pore by PI(4,5)P$_2$[86] stabilization of the active conformation[85]. The present studies suggest that E-Syt2 stabilizes the inactive loose α6 helix state, while VAPA-IRBIT-E-Syt1 stabilizes the active state. Such a mechanism ensures that ANO1 is inactive at the resting state, and when at the active state, its activity is regulated with high fidelity by the cAMP/PKA pathway.

## Methods

### Animal and saliva secretion
All animal protocols have been reviewed and approved by the National Institutes of Health animal use committee (ASP 22-1090) and comply strictly with all National Institutes of Health guidelines. Mice of both sexes were used in these experiments. Generation the IRBIT$^{-/-}$ mice has been previously described[17]. Briefly, exon 2 and flanking sequences were deleted using the required vector. The targeting vector was electroporated into TT2 ES cells and ES clones were injected into 8 cell-stage embryos and transferred into pseudo-pregnant mothers, and male chimeric mice were bred against female C57BL/6 J mice to obtain the IRBIT$^{-/-}$ mice. The IRBIT$^{-/-}$ mice are fertile and were used for breeding. All mice were housed in a controlled environment with regulated temperature and a 12 h dark/light cycle, and they had ad libitum access to both water and food. Sliva secretion was measured from 4-6-month-old mice that were then euthanized, and their parotid glands were removed to prepare isolated acini. Saliva secretion was measured as described before ref. 87. In brief, saliva secretion was stimulated by s.c. injection of 0.125 mg/kg pilocarpine to anesthetized mice (ketamine 60 mg/kg i.m. and xylazine 8 mg/kg i.m.). Saliva was collected from the oral cavity, and the volume was determined by weight.

### Isolation of Parotid acini
Suspensions of isolated acini from the parotid glands of wild-type and IRBIT$^{-/-}$ mice were prepared following the protocol described

previously[17]. Briefly, mice were euthanized by CO$_2$ inhalation, and parotid glands were removed and injected with an ice-cold NaCl-based buffer (Solution A), containing (mM): 140 NaCl, 5 KCl, 1 MgCl$_2$, 10 HEPES (pH 7.4 with NaOH), and 1 CaCl$_2$, supplemented with 0.1% Na$^+$-pyruvate (Sigma-Aldrich), 0.015% soybean trypsin inhibitor (Sigma-Aldrich), and 0.1% bovine serum albumin (Roche). The tissue was finely minced and digested with 0.25 mg/ml collagenase P (Roche) dissolved in solution A and incubated at 37 °C for 6 min under vigorous shaking. After digestion, the acini were washed three times and then suspended in ice-cold solution A without digestive enzymes and were kept on ice until used to measure intracellular Ca$^{2+}$ and Cl$^-$.

### Ca$^{2+}$ measurements
Intracellular Ca$^{2+}$ ([Ca$^{2+}$]$_i$) was measured with Fura-2 using Fura-2 AM (Sigma-Aldrich, 34993). Isolated acini were seeded on coverslips coated with poly-L-lysine (Sigma-Aldrich) and treated with 5 μM Fura-2/AM and 0.05% Pluronic F-127 (Invitrogen, Carlsbad, CA) for 20 min at room temperature. Afterward, the coverslips were inserted into a perfusion system and continuously perfused with a pre-warmed (37 °C) bath solution until the signal stabilized. Subsequently, the perfusate was exchanged either with a calcium-free solution or a bath solution, as specified in the Figs. The Bath solution contained (mM): 140 NaCl, 5 KCl, 1 MgCl$_2$, 10 HEPES (pH 7.4 with NaOH), and either 2 CaCl$_2$ or 0.5 EGTA Ca$^{2+}$-free solution. Images were recorded for at least 3 minutes until the baseline signal was established. The acini were stimulated with carbachol (Sigma-Aldrich) in either a calcium-free solution or a bath solution. Ca$^{2+}$ influx was measured by Ca$^{2+}$ addback by perfusion with a solution containing 2 mM Ca$^{2+}$. Fura-2 fluorescence was captured every 2 s using a TILL imaging system equipped with a 60X objective, employing excitation wavelengths of 340 nm and 380 nm, and collecting emitted light at wavelengths exceeding 510 nm. The images were analyzed with MetaFluor software (Molecular Devices, Sunnyvale, CA), normalized to the baseline signals to calculate the 340/380 ratios, and then averaged to determine release and influx. For calcium oscillations, each trace was utilized to determine the number of responsive cells and fluorescence intensity.

### MQAE fluorescence measurements
Intracellular Cl$^-$ (Cl$^-_{in}$) was evaluated by measuring N-(Ethoxycarbonylmethyl)-6-methoxyquinolinium bromide (MQAE, Biotium, 52011) fluorescence as previously described[17]. Isolated acini attached to poly-L-lysine coated coverslips were incubated with 5 mM MQAE at room temperature for 15 minutes. Afterward, the MQAE dye was removed by continuously perfusing the sample with a warm 37 °C bath solution. The perfusate was switched to a Cl$^-$-free solution by replacing Cl$^-$ in bath solution with NO$_3^-$, and after determining the baseline of MQAE fluorescence, the acini were stimulated with carbachol in the presence and absence of the ANO1 inhibitor Ani9 (5 μM, Cayman Chemical). MQAE fluorescence was recorded every 2 s using the same imaging system and settings described in the Ca$^{2+}$ measurement, with an excitation wavelength of 360 nm and an emission wavelength of 510 nm. All images were measured and analyzed by MetaFluor software.

### Plasmids, mutagenesis and cloning, transfection, and siRNA
ANO1-mCherry (UniProt: Q8BHY3), IRBIT-GFP (GenBank: AB092504.1), E-Syt1 and E-Syt2 (Generously provided by De Camilli lab, Yale University) served as the template for the mutagenesis using QuikChange Lightning Site-Directed Mutagenesis Kit (210518; Agilent Technologies), following the manufacturer's instructions. Primers were obtained from Integrated DNA Technologies. The utilized primers are detailed in Supplementary Table 1. All mutations were validated by sequencing to ensure the presence of the desired mutation. The primers utilized for sequencing are detailed in Supplementary Table 2. FKBP-PI5Ptase and the PtdSer sensor GFP-evt2-2xPH were generously

provided by Dr. Tamas Balla (NICHHD, Bethesda, MD). The PI(4)P (P4M), PI(4,5)P$_2$ (PLCδ-PH), and PtdSer (mRFP-Lact-C2) biosensors were from Addgene.

Cell lines including HEK293T, HEK293T-IRBIT$^{-/-}$ and HEK293T-STIM1$^{-/-}$ (kindly provided by Dr. Trebak[88]), were cultured at 37 °C with 5% CO$_2$ in Dulbecco's modified Eagle's medium (DMEM) supplemented with 10% fetal bovine serum and 1% Penicillin/Streptomycin. Lipofectamine 2000 (11668019; Life Technologies) were used for transfection, following the manufacturer's instructions. Plasmid cDNAs were transfected into cells 18–24 h prior to experiments. If necessary, cells were treated with either 100 nM of scrambled RNA as a control or 100 nM siRNA 48 hours before the transfection of the desired proteins, resulting in a total siRNA treatment duration of 72 hours. All siRNA sequences used are outlined in Supplementary Table 3. For PI(4,5)P$_2$ depletion, cells were additionally transfected with the FRB/FKBP-PI5Ptase system and exposed to 0.2 µM rapamycin for 2 min to induce PI(4,5)P2 depletion before commencing the current measurement. Human submandibular acinar cells (NS-SV-AC), kindly provided by Dr. John Chiorini (National Institute of Dental and Craniofacial Research, NIDCR), were maintained in Defined Keratinocyte Serum-Free Medium (SFM) (Gibco™ 10744019) supplemented with 1% Penicillin-Streptomycin (Pen/Strep). Cells were cultured at 37 °C with 5% CO$_2$ and subcultured at a 1:5 ratio upon reaching 80–90% confluence.

### Generating IRBIT knockout (IRBIT$^{-/-}$) cells

IRBIT$^{-/-}$ cell line 3 was generated by transfecting the Cas9 stable cell line from Genecopoeia (SL502) with three different gRNA for human IRBIT gene in HEK293T cells. The set of 3 gRNA plasmids pCRISPR-SG01 expressing the DNA sequences for gRNA for IRBIT were obtained from GeneCopoeia (cat #HCP291015-SG01-3). The cells were selected in Hygromycin (200 mg/ml) containing media so that untransfected cells died. The live cell colonies were cultured and subcloned in different wells of 24 well plates for 4 weeks. Different cell colonies were expanded in six-well plates, and aliquots were sent for sequencing to confirm the knockout of the IRBIT gene. The cells were harvested, and genomic DNA was extracted using Qiagen DNAeasy blood and tissue kit (cat#69504) and the IRBIT DNA was amplified using primer pairs (870 F and 1400 R and 1100 F and R). The amplified DNA was transformed into DH5 α cells, and eight bacterial colonies from each cell colony were sequenced using the sequencing primers below. The colonies that showed deletions frameshift mutations in both the strands of DNA were expanded, frozen, and used for experiments. The primer paired used for amplifying IRBIT DNA were:1100 F5'- GGCGTGGATGGCCCGCTGGGAGA-3';1100 R 5'-AGCGCTATATCAAAGGTGGAGCCCTG-3'; 870 F 5'-GCAGAGCAGCTTATTAACCC-3' and 1400 R 5'-AATAGGGCTGGGCAGCCAGGGTGGCGC-3'.

### Current recording

Whole-cell voltage-clamp experiments were conducted at room temperature using NS-SV-AC cells, or HEK cells transfected with the desired proteins and maintained in culture media as described above. Cells were detached and seeded onto coverslips 3 h before recording. Transfected cells were distinguished by GFP and/or mCherry fluorescence. Patch pipettes, filled with pipette solution, were pulled from glass capillaries (Warner Instruments) using a PC-100 Narishige vertical puller, yielding resistances ranging from 4 to 6 MΩ. The pipette solution contained (mM): 146 CsCl, 2 MgCl$_2$, 10 sucrose, 8 HEPES, and 5 EGTA, pH 7.4 (with CsOH). The desired free calcium concentration in the pipette solution was adjusted using the Ca-EGTA Calculator (https://somapp.ucdmc.ucdavis.edu/pharmacology/bers/maxchelator/CaEGTA-TS.htm), with settings at 37 °C, 0.165 N, 5 mM EGTA, and pH 7.4. The initial currents were recorded from cells after continuously perfusing the bath solution for at least 1 min to achieve whole-cell configuration, with a capacitance ranging from 15 to 25 pF. Then, the currents were measured from the same cells exposed to 1 µM forskolin (Cayman Chemical, 11018) and 10 µM 3-isobutyl-1-

methylxanthine (IBMX, Cayman Chemical, 13347) in a perfused bath solution. H89 was obtained from (Sigma-Aldrich, 371962). The bath solution contained (mM): 140 NaCl, 5 KCl, 1 MgCl$_2$, 2 CaCl$_2$, 15 Glucose and 10 HEPES (pH 7.4 with NaOH). Current measurements were recorded utilizing an Axopatch 200B capacitor-feedback patch-clamp amplifier connected to a Digidata-1440A Digitizer (Molecular Devices) and were acquired using PClamp 10 software, subsequently analyzed with Clampfit 10 software. A low-pass filter set at 2 kHz was used for the current recording,−100 to +100 mV, with increments of 20 mV from a holding potential set at −60 mV. The currents recorded at +100 mV were used to calculate current density by normalizing to cell size, expressed as pA/pF.

### Co-IP and surface expression of ANO1 by biotinylation

Co-IP and surface expression were assayed with the same cell lysates after biotinylating surface proteins. HEK 293 T cells were transfected with the desired plasmids for 18–24 h, followed by treatment with a vehicle or stimulation with 1–5 µM Forskolin and 10–100 µM IBMX in the bath solution, as indicated, for 10 min at 37 °C. The antibodies used for total, Co-IP, and surface experiments were obtained from: Monoclonal anti-Flag (Sigma, F3165) 1:2000 dilution; Polyclonal anti-GFP (Life Technologies, A11122) 1:3000 dilution; Monoclonal anti-MYC (Cell Signaling Inc., 2276) 1:1000 dilution; Polyclonal anti-ANO1 (Abcam, 53212) 1:1000 dilution; Polyclonal anti-GAPDH (Cell Signaling Inc.,2118) 1:2000 dilution; Monoclonal anti-beta actin (Sigma, A3854); 1:3000 dilution; Polyclonal anti-E-Syt1 (Sigma, HPA016858) 1:1000 dilution; Polyclonal anti-E-Syt2 (Sigma, HPA002132) 1:1000 dilution; Polyclonal anti-AC8 (Invitrogen PA5-72589) 1:1000 dilution; Polyclonal anti-AC6 (Proteintech, 14616-1-AP) 1:1000 dilution.

Total and surface expression of ANO1 and desired proteins were prepared according to the previously established protocol[38]. Briefly, after rinsing the cells with PBS, they were incubated in a cold biotinylation solution (0.5 mg/mL EZ-LINK™ Sulfo-NHS-SS-biotin, Thermo Fisher Scientific, 21331) on ice for 30 min. The biotinylation process was stopped by washing the cells three times with cold PBS and then incubating them with a 50 mM glycine solution for 10 minutes. The proteins were extracted by extracting the cells in ice-cold lysis buffer (20 mM Tris, 150 mM NaCl, 2 mM EDTA, 1% Triton X-100, and a protease inhibitors cocktail (Roche, 11836170001) using a Potter homogenizer (GEX 130 PB at 20% amplitude). Afterward, the mixture was left to incubate on ice for 15 minutes and then collected by centrifugation at 4 °C. The biotinylated proteins were collected by overnight incubation with Biosystem's Magnetic Beads Neutravidin (# 29204; Thermo Fisher Scientific) at 4 °C. The captured proteins were eluted by heating in an SDS sample buffer at 56 °C for 20 min, separated by SDS/PAGE, and transferred onto a membrane for immunoblotting. The membrane was probed using specific antibodies against Myc (Myc-ANO1), Flag (Flag-IRBIT), GPF (GFP-VAPA), and GAPDH. ImageJ was utilized to quantify the band intensity of all blots. All values were normalized to the surface ANO1/input ANO1 or surface ANO1/input GAPDH ratio.

### TIRF and confocal microscopy and FRET measurements

Proteins at and close to the plasma membrane (PM) were determined by TIRF microscopy, and protein interaction and proximity was analyzed by FRET microscopy with fluorescently tagged proteins. HEK293 cells plated on glass-bottom dishes (MatTek Corporation, Ashland, MA) were transfected with the desired plasmids, and incubated for 18–24 h. The initial images were captured after cells were incubated in a pre-warmed bath solution for at least 1 minute until the signal stabilized. Then, 2.5 µM FSK and 50 µM IBMX were added to the solution, and after an additional 5 min of incubation, images were acquired again. TIRF imaging was performed as described before[38] using a Nikon NIS-Elements system coupled with the Nikon Eclipse Ti. This setup includes the PFS (Perfect Focus System) for autofocus. In addition, it

features a Nikon N-Storm, Andor iXon Ultra Camera with EMCCD Sensor, D-Eclipse C1, and a 60x TIRF objective lens (Nikon) with specifications of 1.45 NA Oil immersion, infinity/0.10-0.22 DIC H. Images were adjusted for background fluorescence if needed. Because many treatments increased both the number of puncta and fluorescence intensity of existing puncta, the change of fluorescence at the TIRF field was analyzed by measuring fluorescence intensity that was then normalized. The normalization of puncta intensity of the selected cell area was analyzed using the NIS-Analysis software provided by Nikon Imaging.

Confocal images were captured using a Yokogawa CSU-X1 with Filter Wheel Control and shutter, a Photometrics® Evolve Delta camera, and Olympus UPlanSApo objectives (60X, 1.35 NA Oil immersion, infinity/0.13-0.19/FN22).

FRET measurements were performed using MetaMorph software paired with the Olympus IX81 Microscope and Olympus IX2-UCB, with power conditioning provided by a Tripp-Lite Line Conditioner LC2400 connected to the microscope. Epifluorescence and FRET measurements utilized the Air-Therm ATX-H and CoolLED pE-300 LED Illuminator, as well as the ASI MS-2000, and Vortran Laser Technology, Inc. Diode Module Stradus® Control Box with CDRH ON/OFF key switch, featuring 405, 445, 488 nm, 515, 561, and 639 nm lasers attached to a Triggerscope V-3B and Laser Aperture. Data acquisition and processing were conducted on a pixel-by-pixel basis, employing a two-step FRET efficiency calculation protocol as previously described[38]. The results were then exported to GraphPad Prism 10 for statistical analysis and presentation.

### Prediction of PKA phosphorylation sties in Anoctamin 1

The phosphorylation sites of ANO1 (UniProt: Q8BHY3) and its isoform a (GenBank: BBE32313.1) at serine (Ser), threonine (Thr), and tyrosine (Tyr) residues were predicted using the Group-based Prediction System (GPS) 6.0 with a light gradient boosting machine (https://gps.biocuckoo.cn/online_LGBM.php), specifically for PKA-specific phosphorylation sites. The threshold was settled at medium, and the cutoff was 0.1447. The details of the prediction results are shown in Fig. S1D.

### Statistics

The experiments were replicated a minimum of three times, and all data are presented as mean ± standard error of the mean (SEM). Each dot within the columns represents an individual cell that was analyzed. Statistical significance was assessed using unpaired Student's $t$ test, with GraphPad Prism 10 software. The $p$-values are provided in the Figures, and those below 0.05 are considered statistically significant.

### Reporting summary

Further information on research design is available in the Nature Portfolio Reporting Summary linked to this article.

## Data availability

All data and materials generated as part of these studies in our lab are available on reasonable request and upon satisfying NIH guidelines. Source data are provided in this paper.

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

## Acknowledgements

We thank Dr. Tim Levine (UCL Institute of Ophthalmology, London) for suggesting the IRBIT FFAT motif. Dr. Pietro De Camilli (Yale University) for generously providing the plasmids coding for all E-Syts, Dr. Tamas Balla (NICHD/NIH) for GFP-evt2-2XPH, Dr. John Scott (Washington University) for AKAP11, Dr. Mark Dell'Acqua (University of Colorado) for AKAP79-mTurqoise2 and AKAP-EYFP and the NIDCR imaging core for the help with image acquisition and analysis. We thank Dr. Mohamed Trebak (University of Pittsburgh) for the STIM1$^{-/-}$ cells. This work was funded by an intramural NIH grant, NIH/NIDCR DE000735-13. Phosphoproteins analysis by MsS was done at the NIDCR Mass Spectrometry Facility and supported by grant ZIA DE000751.

## Author contributions

W.-Y.L., W.Y.C., S.P., A.M.A., B.L., and M.A. performed experiments and analyzed data; S.M. conceptualized and directed the study, acquired support, analyzed data, and drafted the manuscript with contributions from all authors.

## Funding

## Competing interests

The authors declare no competing interests.
