## [Transparent Peer Review file · Nature Communications]

Multiple cAMP/PKA Complexes at the STIM1 ER/PM Junction, Specified by E-Syt1 and E-Syt2, Reciprocally Gates ANO1 (TMEM16A) via Ca²⁺

Corresponding Author: Dr Shmuel Muallem

Version 0:

Reviewer comments:

Reviewer #1

(Remarks to the Author)

ANO1 is a calcium-dependent chloride channel. The physiological significance of ANO1 has been established. However, it remains to be defined how the activity of ANO1 is regulated. In this manuscript, Lin et al have systematically analyzed the expression and modification of ANO1 and its subcellular localization. One of the main conclusions of the work is that ANO1 is involved in a large protein complex at the ER-PM contact sites. The interaction of ANO1 with VAPA and its phosphorylation by PKA are important for its expression. At the resting state, E-Syt2 plays a role in its inhibition. Upon Calcium release, E-SYt1/STIM are required for its activation. Overall, this is an interesting and well-written paper, the quality of the data is very high.

In a revised version of the paper, the authors should address the following points:

- 1, Can the author show the subcellular images confirming that ANO1 is colocalized with VAPA, but not VAPB?
- 2, Can author add a brief discussion on why and how E-Syt1/2 switch the activation state of ANO1? Do you have idea which domain of E-Syt1/2 is involved in this interaction?
- 3, It is not clear that how the phosphorylation of ANO1 regulates its expression level.
- 4, Does the lipid transfer by E-Syts is required for the ANO1 regulation?

Reviewer #2

(Remarks to the Author)

In this work, Muallem's group conducted a comprehensive series of experiments to test a novel regulation mechanism of the TMEM16A calcium-activated chloride channel. The authors propose that a cAMP/PKA-dependent phosphorylation modulates the expression of TMEM16A at the plasma membrane and its Ca sensitivity. Moreover, the authors offer that this regulatory mechanism occurs at the STIM1-rich junction of the plasma membrane with the endoplasmic reticulum. The authors put forward the phosphorylation of S673 residue after the formation of a large complex comprising ANO1-VAPA-IRBIT-E-Syt1-AC8-AKAP5-PKA to enhance the expression of ANO1 at the plasma membrane and Ca sensitivity. Furthermore, they suggest that the complex ANO1-IRBIT-E-Syt2-AC6-AKAP11-PKA does the opposite by phosphorylating S221. They conducted a proof-of-concept experiment that deleted the IRBIT expression in parotid acinar cells, which impaired fluid secretion, highlighting the probable significance of their findings. While the general proposal is intriguing and potentially holds significant physiological relevance, the current manuscript is challenging to navigate. Moreover, several issues with the data detract from the overall enthusiasm for the research.

In the following paragraphs, a list of significant issues is presented.

1. While the abstract, introduction and discussion are well-crafted and reader-friendly, the result section is challenging due to

the overwhelming amount of information in each figure. The inclusion of several Figures with 16 panels and, in at least one case, 25 panels is excessive. It is crucial for the authors to establish a clear storyline that connects all the findings, as this would greatly enhance the readers' understanding of the research. For instance, the diagram displayed in the supplementary material could be placed early in the results to help readers grasp the hypothesis, the experimental approach used to address the hypothesis, and the results' outline and significance. Prioritizing key findings and presenting them in a more structured and concise manner would also greatly improve the clarity and impact of the manuscript.

2. According to the author's idea, the channel transits from a desensitised (low Ca affinity) to a sensitised (high Ca affinity) state upon cell stimulation with a PLC-dependent Ca mobilising agonist. According to this idea, in the absence of the cAMP/PKA cascade, the channel will remain desensitised with a low Ca sensitivity and cannot respond to the agonist. This idea contradicts previous observations showing that purified TMEM16A respond to Ca with an apparent Ca affinity of 210 nM when reconstituted in liposomes (<http://www.pnas.org/cgi/doi/10.1073/pnas.1312014110>) in the absence of ATP, calmodulin, etc. Furthermore, intact HEK cells have endogenous IRBIT, presumably allowing the channel to reach the sensitised state upon stimulation with carbachol. So, how much would alter the expected response or even the normal physiology of the cells by the overexpression of IRBIT?

3. The MQAE data presented in Figure 1 E-H need further explanation. Unless the authors show that maximum and minimum fluorescence signal values are the same, the Fura ratios alone cannot be equated to equal intracellular Ca concentrations. The G plot indicates that in IRBIT^{-/-} acini, 100 μ M carbachol induces about 25% of the control response. Yet, the F plot shows that the MQAE fluorescence at the end of the IRBIT^{-/-} carbachol stimulation is closer to 50% of that generated by the WT-stimulated acini. Furthermore, the H plot shows that the apparent sensitivity of the acini to carbachol decreases in the absence of IRBIT. Is there an explanation for this result? Could an alternate idea explain this observation? The authors should perform electrophysiological measurements to demonstrate that the endogenous CaCC decreased by IRBIT ablation.

4. The electrophysiology seems less credible. Phosphorylation plays a critical role in the author's hypothesis. However, the cytosolic solution used to record the macroscopic currents lacked ATP, which is required to support the phosphorylation of TMEM16A, as Leblanc's group has shown. Also, Suh's group recently showed that ATP decreased the current through the a and ac variants of TMEM16A. So, the experiments should be done in the presence of ATP to claim a phosphorylation-dependent mechanism. The authors showed IV curves collected from cells dialysed with 0.3, 1, and 10 μ M Ca. Its linearity is a hallmark of the IV curves from cells dialysed with 10 μ M Ca; this behaviour does not seem to be the case in this study (Figure 5B). In some cases, the rectification seems more pronounced with 10 μ M than with 0.3 μ M Ca. If the small current recorded with 0.3 μ M Ca is contaminated by leak + linear currents, that would partly explain this behaviour. Considering the high Ca and symmetrical chloride concentration, the IV curve should be linear with 10 μ M Ca. Furthermore, the magnitude of the whole cell current seems relatively small. The authors said that the range of cell capacitance values was 15-25 pF. Considering a 20-pF cell and its average current density at +100 mV, all their cells have a current smaller than 0.5 nA with 10 μ M Ca. This isn't easy to understand since some authors report current magnitudes near 0.5 nA when recording TMEM16A from inside-out patches (<https://doi.org/10.1085/jgp.201611650>, <https://doi.org/10.1085/jgp.201611651>). Additionally, the reported apparent Km values for the WT TMEM16A at +100 mV are higher than 0.5 μ M Ca. This value is like those reported at -40 or -80 mV (<https://doi.org/10.1371/journal.pone.0086734>; <https://doi.org/10.1085/jgp.201611650>). Finally, because these results could be physiologically relevant at the resting membrane potential of cells containing 30-50 mM chloride in their cytoplasm, the authors should perform their electrophysiologic analysis at potential near the resting membrane potential and using a cytosolic solution containing around 40 mM chloride.

5. There seems to be a misinterpretation regarding the expression versus localisation of proteins in the TIRF experiments. While the TIRF data show membrane localisation of ANO1, this does not imply a change in its expression levels, as suggested in the manuscript. The author's method said, "The normalization of puncta intensity of the selected cell area was analysed using the NIS-Analysis software provided by Nikon". How the areas were selected and whether the puncta were individually analysed is unknown. Also, in lane 323, the authors claim, "TIRF measurements showed that inhibition of ANO1 by E-Syt2 was in part due to reduced ANO1 surface expression, even when increased by IRBIT (Figure 5C)". Again, getting the surface expression of ANO1 from the TIRF data is problematic because the experimentalist is only sampling a small area of the membrane. A more punctuated pattern can be observed within the same location or an even smaller amount of protein. However, it is problematic to translate such a piece of information into a global increase or decrease in protein expression. To support that claim of increased expression, a Western blot analysis using membrane protein isolated from biotinylated membranes should be performed to confirm any changes in protein levels. TIRF analysis should include a membrane marker to ensure accurate identification of membrane localisation. Figure 2G shows ANO1 fluorescence data without and with overexpressed IRBIT. How do we know that IRBIT is present in the cells shown in the right panels? The puncta and the whole cell current at +100 mV increased, but such an increase seems less spectacular than the surface increase of ANO1 (panel 2H).

6. The assumed interaction between TMEM16A and different cytosolic proteins is another critical issue. In some cases, this assumption is supported by the FRET data. However, co-immunoprecipitation assays should be performed to confirm that all those proteins interact within the native cells. This is particularly relevant because all current experiments used a heterologous expression system that overexpressed the proteins.

7. Several grammatical errors in the manuscript need to be fixed. Some of these errors are important. For example, in line 131, it is stated that the work was done using the TMEM16A variant a. However, the numbering indicates that the variant used in this work was the ac. For example, residue 673 corresponds to S673 in the ac variant. However, the homologous residue in the variant a is S669. Another example is in line 438; ANO8 and ANO6 were mistyped instead of AC8 and AC6. Also, there are no columns in Figure 3A. Even though some of these errors may have had little incidence in the final interpretation, they can confuse new readers and no experts in the field.

8. The organisation of the figures is unclear, making it difficult to follow the main concepts. Implementing a colour-coded scheme for the graphs would improve the readability and help track the results more quickly. Maintain uniformity in the graph labels by standardising the font size and style across all figures. Keep plot size consistent and balance the distribution of elements within the figures to improve clarity and visual consistency.

9. The phosphorylation of residue S673 is another critical issue. Ko et al. showed that residue S673 is the target of CaMKII (<https://doi.org/10.1073/pnas.2014520117>). Accordingly, they showed that KN-62, a blocker of CaMKII, increased TMEM16A current, whereas KT5720, an inhibitor of PKA, had no effect. And yet, in this work, the authors claim that PKA phosphorylates S673. Furthermore, according to Ko's report, the ANO1 S673D current is more minor than WT's; however, in this work, the ANO1 S673D current is more significant than that of the WT current. These issues need to be resolved.

10. Line 155: TMEM16A is regulated but not activated by IRBIT.

11. Lines 181-184: How much PIP2 was depleted? How long were the rapamycin stimuli? The current magnitudes are very small!

12. Fig 5: It seems that E-Syt2 does not inhibit the current. Instead, the Ca sensitivity is affected. The current is smaller because it is less sensitive to Ca.

13. The Hill Equation cannot describe the data in Fig 5H. It is more complex than that. Therefore, the Kms in Fig 5J are not credible.

14. In line 344, the authors suggest that "it appears that the various combinations of E-Syt2 and IRBIT make complexes with different ACs to differentially determine the surface expression level of ANO1 and its activation by Ca²⁺". However, in addition to the author's suggestion, the open probability could also be affected more than the number of channels. Such affection would also help to explain the result and should be discussed at the very least.

15. The labels of the columns should be clear because there are multiple treatments. It is confusing to see a "+" before the label because it is not clear what the interpretation should be. Also, if possible, a colour-coded pattern should be followed throughout the manuscript figures.

Reviewer #3

(Remarks to the Author)

The study reveals a new function of STIM1 in the assembly and transduction of the cAMP signaling pathway and elucidate a novel mechanism regulating ANO1 surface expression and Ca²⁺ gating. I have some relatively minor comments.

INTRODUCTION

1. line 50: "ANO1 is a decision-making channel" - what is a decision-making channel? Clarify or delete

2. Overall, the introduction is far too long - please condense to the essentials

METHODS and DATA

1. It is not clear why IRBIT did not increase the current beyond the current measured with ANO1 and VAPA if there is synergy, as the authors say - can you clarify?

2. The AC experiments are not clear-AC6 and AC8 are both essential (Lines 292-304) - which goes along with the complexity described in lines 410-436.

3. Line 436: compartmentalization of ACs is not as well studied as that of AKAPs. It should be mentioned here. I am glad the authors studied AKAPs, however, see below

4. It is not clear if the authors are studying PKA-I or -II (R1a or R2a). By the AKAPs, it is PKA-II but it needs to be clarified and experimentally confirmed

Figures

...are hard to see...

Version 1:

Reviewer comments:

Reviewer #1

(Remarks to the Author)

The authors addressed all my concerns with new experiments and explanations. I do not have further major request.

Reviewer #2

(Remarks to the Author)

I appreciate the author's efforts to address my concerns. Many of them were satisfactorily answered. However, I still have concerns with the following issues.

1. The authors claim that IRBIT regulates TMEM16A affinity. How can this happen? The Ca affinity is an intrinsic property of TMEM16A that depends on the arrangements of acidic residues forming a pocket holding 2 Ca ions. Enhancing or decreasing the affinity would require altering the pocket structure. How can IRBIT accomplish this affinity change? The authors should indicate in the main text that the change in the apparent K, estimated using the Hill Equation, suggests that the affinity may vary. Also add a line in the methods indicating how the apparent K_d was estimated.

2. I think the authors are overextending their conclusions. In their Discussion (Lines 735-746), the authors described their hypothesis and explained their observations on how phosphorylation would change channel density and apparent affinity.

For example, they said “Cell stimulation that releases Ca²⁺ from the ER results in STIM1 translocation along with E-Syt1 to ER/PM junctions. The STIM1 and E-Syt1 translocation increases junctional PI(4)P and PI(4,5)P₂ levels, facilitating formation of the ANO1-VANZ-E-Syt1 complex and its association with the AC8-AKAP5-PKA complex, leading to the phosphorylation of ANO1 Serine 673. As a result, the reserve ANO1 channels close to the PM are inserted into the PM, and ANO1 adopts a sensitized conformation with high Ca²⁺ affinity, becoming fully activated at 1 μM Ca”. a) Such an orderly string of interactions requires time to occur and reach a steady state. It has been previously shown that the chloride current follows the Ca signal with a very short delay (DOI: 10.1113/jphysiol.2001.013453). The onset of the chloride current has a time constant smaller than 20 ms in parotid acinar cells upon a Ca jump. I wonder whether the interactions and channel translocation described here can occur on this time scale to support the proposed hypothesis. b) The so-called reserve of ANO1 channels must be nearly equal to that in the plasma membrane to increase ~2-fold V_{max}. Is there any evidence to back up this idea? c) The apparent affinity of ANO1 has been determined from inside-out patches. Under this experimental condition ANO1 displays high affinity (<https://doi.org/10.1085/jgp.201611650>, doi:10.1371/journal.pone.0086734). The apparent affinity is like that determined from whole acinar cells (see cite above) and higher than those here described. However, this experimental condition does not support the mechanism proposed by the authors. How can the authors’ hypothesis be reconciled with those observations? d) the 5OYB structure, which has 2 Ca bound, have a non-conductive pore because the pore diameter is too narrow to allow chloride permeation. Also, molecular dynamics simulations (<https://doi.org/10.1038/s42003-021-01782-2>) indicate that the pore of the 5OYB structure is not conductive. In this work, it was shown that PIP₂ ligation induced pore dilation, which allowed chloride permeation. Therefore, the idea that “binding of a second Ca²⁺ ion, and stabilization of the active conformation” needs to be revised.

3. The data with ATP in the pipette solution should be in the main text because a phosphorylation mechanism seems to be central for explaining the data. Is the S673A mutant insensitive to IRBIT? When ATP is added to the pipette solution to dialyze the cytosol, I doubted that ATP would be compartmentalized. Please clarify what compartmentalized ATP means.

4. Please clarify the legend of the Supplementary Figure 2 (F-H), which shows the total ANO1 currents of Figures 2J-L.

Reviewer #3

(Remarks to the Author)

The authors have addressed most comments in a way that made the manuscript much stronger. The co-IP experiments are convincing. The presentation is clearer.

Response to reviewers' comments:

We greatly appreciate the reviewers' positive evaluation of the manuscript and their constructive comments. In response to the comments, we conducted several additional experiments that required the inclusion of an additional Figure, now labeled as Figure 9. These new experiments are listed below. We also extensively revised the manuscript by transferring over 50% of the panels to the supplementary material that now includes 16 Figures. The new Figures are listed below in the order of their appearance in the text:

1. Fig. 1E: Measured muscarinic-stimulated salivary glands fluid secretion in the IRBIT^{-/-} mice to increase the physiological relevance of the finding. (see also below measurement with the native ANO1 in NS-SV-AC cells).
2. Supplementary Fig. 1A: Demonstrating linear I/V at high Ca²⁺ when ANO1 is expressed at high level and the weak effect of forskolin/IBMX under these conditions.
3. Supplementary Fig. 1E: Demonstrating that the effect of IRBIT and F/I is independent of cytoplasmic Cl⁻ concentration.
4. Supplementary Fig. 1I: Confocal images of E-Syt1-ANO1 and E-Syt2-ANO1 co-localization
5. Supplementary Fig. 2B: Demonstrating that including ATP in the pipette solution has no effect on activation of ANO1 by IRBIT and by stimulation with forskolin.
6. Figure 2L: Including a guiding model of the proteins and the complexes to be analyzed in Figs. 3-9.
7. Supplementary Fig. 3B-C: Comparing the interaction of ANO1 with VAPA and VAPB by FRET.
8. Fig. 4D: Effect of F/I stimulation on E-Syt1-ANO1 Co-IP and on ANO1 surface expression by biotinylation.
9. Fig. 5B: Effect of F/I stimulation on E-Syt2-ANO1 Co-IP and on ANO1 surface expression by biotinylation.
10. Supplementary Figs. 14A-B: Analysis of E-Syts, their SMP lipid transfer mutants and their effect of ANO1 localization at the TIRF field.
11. Supplementary Figs. 14C-J:
 - a) Analysis of the effect of E-Syt1 and E-Syt2, their SMP mutants and depletion of E-Syt1 and E-Syt2 on plasma membrane PI(4)P, PI(4,5)P₂ and PtdSer.
 - b) Demonstrating that the steady state level of the lipids is determined by the reciprocal but concomitant action of E-Syt1 and E-Syt2.
 - c) Showing that E-Syt2 dominates over E-Syt1 in determining the junctional lipid composition.
12. Supplementary Figs. 14K: Demonstrating the outward rectifying, time and voltage dependence of ANO1 expressed at low level and the native ANO1 at 10 μM Ca²⁺.
13. Figs. 9A-F: Demonstrating the reciprocal effects of the E-Syt1 and E-Syt2 SMP lipid mutants compared to E-Syt1 and E-Syt2 on activation of ANO1 by Ca²⁺.
14. Fig. 9G: Demonstrating that depletion of E-Syt1 inhibits the increased current by the E-Syt2 SMP mutant E-Syt2(V197W/I337W).

15. Fig. 9H: Demonstrating that deleting or mutating the PI(4,5)P₂ binding sites of E-Syt1 and E-Syt2 inhibited and partially reversed their function and eliminated the effect of F/I.

16. Supplementary Fig. 15: Western blot analysis of the *native* ANO1, VAPA, IRBIT, E-Syt1, E-Syt2, AC8 and AC6 proteins in the immortalized human salivary gland acinar NS-SV-AC cells.

17. Figs. 9I-J: Demonstrating the activity of the native ANO1 current in NS-SV-AC cells.

18. Figs. 9K-M: Effect of depletion of VAPA, E-Syt1, E-Syt2, IRBIT, STIM1, AC8, AC6, AKAP3, AKAP5 and AKAP11 of the current of the *native* ANO1.

Due to the large amount of additional data, the extensive transfer of panels to the supplement and the inclusion of the new Figure 9, the text and Figures legend have been extensively revised. It was not possible to submit a text with all changes marked using track changes. Instead, all text and figure legend changes are highlighted in yellow.

We hope our responses to the reviewers' comments and the additional information provided meets your expectations. Furthermore, we trust that the revised version of the manuscript addresses all your concerns and is now acceptable for publication.

Reviewer 1:

ANO1 is a calcium-dependent chloride channel. The physiological significance of ANO1 has been established. However, it remains to be defined how the activity of ANO1 is regulated. In this manuscript, Lin et al have systematically analyzed the expression and modification of ANO1 and its subcellular localization. One of the main conclusions of the work is that ANO1 is involved in a large protein complex at the ER-PM contact sites. The interaction of ANO1 with VAPA and its phosphorylation by PKA are important for its expression. At the resting state, E-Syt2 plays a role in its inhibition. Upon Calcium release, E-SYT1/STIM are required for its activation. Overall, this is an interesting and well-written paper, the quality of the data is very high.

Response: We greatly appreciate the positive and constructive comments. Testing the role of the lipid transfer SMP domain has substantially enhanced the significance of the manuscript and strengthened the validity of our conclusions.

1. Can the author show the subcellular images confirming that ANO1 is colocalized with VAPA, but not VAPB?

Response: Please note that to an extent VAPB should localize with ANO1 since VAPB slightly affects ANO1 function (Supplementary Fig. 3). However, the new FRET experiments in panel C show a significantly weaker interaction between VAPB and ANO1 compared to VAPA. This interaction is not affected by mutating the ANO1 FFAT motif.

2. Can author add a brief discussion on why and how E-Syt1/2 switch the activation state of ANO1? Do you have idea which domain of E-Syt1/2 is involved in this interaction?

Response: We think that their localization at the ER/PM junctions is likely different. Previous studies showed that the third C2C domains of the E-Syts mediate interaction with plasma membrane lipids, such as PI(4)P, PI(4,5)P₂. Furthermore, electron microscopy (EM) studies indicated that the localization of three E-Syts at the junctions appear differs (see PMID: 23791178). Accordingly, we now show that deletion of the third C2C domains in E-Syt1 and E-Syt2 eliminate their function.

3. It is not clear that how the phosphorylation of ANO1 regulates its expression level.

Response: After demonstrating that surface protein level determined by biotinylation and TIRF microscopy give consistent results, we used TIRF to assess the effect of phosphorylation mutants on surface ANO1. Figure 2G shows that S673A/D mutants do not affect the basal expression of ANO1. Similarly, Figure 5H shows that the S221A mutation had no effect on basal ANO1 expression while the S221D mutation reduced basal ANO1 expression, mimicking the effects of cAMP pathway stimulation in cells expressing E-Syt2. However, we have not examined the effect of these mutations on protein trafficking or insertion into the plasma membrane.

4. Does the lipid transfer by E-Syts is required for the ANO1 regulation?

Response: Thank you very much for this suggestion. Please refer to the new supplementary Figure 14 (lipid transfer) and Fig. 9 for the remarkable role of the lipids in the function of the E-Syts. In fact, effects of the E-Syts on the lipids at the junctions have emerged as the major mechanism by which they regulate ANO1 function.

Reviewer 2:

In this work, Muallem's group conducted a comprehensive series of experiments to test a novel regulation mechanism of the TMEM16A calcium-activated chloride channel. The authors propose that a cAMP/PKA-dependent phosphorylation modulates the expression of TMEM16A at the plasma membrane and its Ca sensitivity. Moreover, the authors offer that this regulatory mechanism occurs at the STIM1-rich junction of the plasma membrane with the endoplasmic reticulum. The authors put forward the phosphorylation of S673 residue after the formation of a large complex comprising ANO1-VAPA-IRBIT-E-Syt1-AC8-AKAP5-PKA to enhance the expression of ANO1 at the plasma membrane and Ca sensitivity. Furthermore, they suggest that the complex ANO1-IRBIT-ESyt2-AC6-AKAP11-PKA does the opposite by phosphorylating S221. They conducted a proof-of-concept experiment that deleted the IRBIT expression in parotid acinar cells, which impaired fluid secretion, highlighting the probable significance of their findings.

While the general proposal is intriguing and potentially holds significant physiological relevance, the current manuscript is challenging to navigate. Moreover, several issues with the data detract from the overall enthusiasm for the research.

Response: We thank you for your fair and constructive criticism and for indicating that the "proposal is intriguing and potentially holds significant physiological relevance". We have made every effort to fully address all of your concerns. We believe that demonstrating the central role of lipid transfer by the SMP domain in the role of the E-Syts, as suggested by reviewer 1, has significantly enhanced the significance of our studies and add a novel aspect to the regulation of ANO1 function reported in the present work. Moreover, to further enhance the physiological significance of our findings, we now show that native ANO1 in the immortalized human salivary gland cell line NS-SV-AC is regulated by two complexes that sensitize and desensitize ANO1 function.

1. While the abstract, introduction and discussion are well-crafted and reader-friendly, the result section is challenging due to the overwhelming amount of information in each figure. The inclusion of several Figures with 16 panels and, in at least one case, 25 panels is excessive. It is crucial for the authors to establish a clear storyline that connects all the findings, as this would greatly enhance the readers' understanding of the research. For instance, the diagram displayed in the supplementary material could be placed early in the results to help readers grasp the hypothesis, the experimental approach used to address the hypothesis, and the results' outline and significance. Prioritizing key findings and presenting them in a more structured and concise manner would also greatly improve the clarity and impact of the manuscript.

Response: We greatly appreciate the suggestions that have clearly helped simplify the presentation. We have now made extensive use of the supplement, moving many of the supporting panels to the supplement which now includes 16 Figures. As suggested, we have added a panel (Figure 2L) to introduce the reader to the complexes to be tested. We refer to the model before describing the roles of VAPA, E-Syt1, E-Syt2, the ACs and AKAPs.

2. According to the author's idea, the channel transits from a desensitised (low Ca affinity) to a sensitised (high Ca affinity) state upon cell stimulation with a PLC-dependent Ca mobilising agonist. According to this idea, in the absence of the cAMP/PKA cascade, the channel will remain desensitised with a low Ca sensitivity and cannot respond to the agonist. This idea contradicts previous observations showing that purified TMEM16A respond to Ca with an apparent Ca affinity of 210 nM when reconstituted in liposomes (<http://www.pnas.org/cgi/doi/10.1073/pnas.1312014110>) in the absence of ATP, calmodulin, etc. Furthermore, intact HEK cells have endogenous IRBIT, presumably allowing the channel to reach the sensitised state upon stimulation with carbachol. So, how much would alter the expected response or even the normal physiology of the cells by the overexpression of IRBIT?

Response: Thank you for your thoughtful question. Indeed, ANO1 has intrinsic activity that is activated by Ca^{2+} binding to its Ca^{2+} binding sites. The purified protein is expected to be activated by Ca^{2+} , but this does not imply that its activity cannot be further modulated by phosphorylation of specific serine residues. The conformation of the purified and reconstituted protein may resemble that of the sensitized ANO1. Our findings suggest that *in vivo* ANO1 exists in different conformations that are determined by the complexes it forms with the cAMP/PKA pathway and the serine residues that are phosphorylated. These findings do not conflict with the activity observed for the unphysiological behavior of the purified and reconstituted ANO1 in an artificial phospholipid environment. Indeed, the new results shown in Figure 9J-L, involving the native ANO1, provide clear evidence that this is the case, showing that in the cell, ANO1 is regulated by the tethers and the cAMP/PKA pathway in the same manner as the ANO1 expressed in HEK cells. We have clarified this in the text.

3. The MQAE data presented in Figure 1 E-H need further explanation. Unless the authors show that maximum and minimum fluorescence signal values are the same, the Fura ratios alone cannot be equated to equal intracellular Ca concentrations. The G plot indicates that in IRBIT^{-/-} acini, 100 μM carbachol induces about 25% of the control response. Yet, the F plot shows that the MQAE fluorescence at the end of the IRBIT^{-/-} carbachol stimulation is closer to 50% of that generated by the WT-stimulated acini. Furthermore, the H plot shows that the apparent sensitivity of the acini to carbachol decreases in the absence of IRBIT. Is there an explanation for this result? Could an alternate idea explain this observation? The authors should perform electrophysiological measurements to demonstrate that the endogenous CaCC decreased by IRBIT ablation.

Response: Thank you for noticing this. We apologize for not presenting the MQAE results as F/F_0 , as originally determined. Presenting the fura2 ratio as 340/380 and the MQAE results as F/F_0 make the measurements independent of dye loading and allows for the comparison between wild-type and IRBIT^{-/-} cells. Moreover, your comment made us realize that actually there is an excellent agreement between the results obtained in HEK cells and the acini. As shown in Figure 4E, in IRBIT^{-/-} HEK cells, ANO1 activity is 2.58 pA/pF, which increased to 6.32 pA/pF by re-expression of IRBIT, representing a 2.45 ratio. In the MQAE experiments, the Cl^- flux V_{max} in the IRBIT^{-/-} acini was 1.57 F/F_0 , while in wild-type acini, it was 4.28 F/F_0 , corresponding to a 2.72-fold increase. Hence, the two sets of data support each other. This is now indicated in the text in relation to Figure 4E.

4. The electrophysiology seems less credible. Phosphorylation plays a critical role in the author's hypothesis. However, the cytosolic solution used to record the macroscopic currents lacked ATP, which is required to support the phosphorylation of TMEM16A, as Leblanc's group has shown. Also,

Suh's group recently showed that ATP decreased the current through the α and αc variants of TMEM16A. So, the experiments should be done in the presence of ATP to claim a phosphorylation-dependent mechanism.

Response: We are very much aware of these data. However, to minimize spontaneous phosphorylation that can mask some of the effects of the ACs and AKAPs, we decided to rely on the compartmentalized ATP. To this end, we show in the new supplementary Fig. 2B that including ATP in the pipette solution had no effect on the response to IRBIT and F/I stimulation.

The authors showed IV curves collected from cells dialysed with 0.3, 1, and 10 μM Ca. Its linearity is a hallmark of the IV curves from cells dialysed with 10 μM Ca; this behaviour does not seem to be the case in this study (Figure 5B). In some cases, the rectification seems more pronounced with 10 μM than with 0.3 μM Ca. If the small current recorded with 0.3 μM Ca is contaminated by leak + linear currents, that would partly explain this behaviour. Considering the high Ca and symmetrical chloride concentration, the IV curve should be linear with 10 μM Ca. Furthermore, the magnitude of the whole cell current seems relatively small. The authors said that the range of cell capacitance values was 15-25 pF. Considering a 20-pF cell and its average current density at +100 mV, all their cells have a current smaller than 0.5 nA with 10 μM Ca. This isn't easy to understand since some authors report current magnitudes near 0.5 nA when recording TMEM16A from inside-out patches (<https://doi.org/10.1085/jgp.201611650>, <https://doi.org/10.1085/jgp.201611651>). Additionally, the reported apparent K_m values for the WT TMEM16A at +100 mV are higher than 0.5 μM Ca. This value is like those reported at -40 or -80 mV (<https://doi.org/10.1371/journal.pone.0086734>; <https://doi.org/10.1085/jgp.201611650>).

Response: The magnitude of the current is dependent on the ANO1 expression level, and we used low expression for several reasons. As shown in the newly added supplementary Figs. 1A-B, measurements of ANO1 current in cells transfected with high level of ANO1 showed that the current is similar in magnitude to that reported by others. However, although the current was increased by IRBIT, stimulation with very high concentration of 10 μM forskolin and 100 μM IBMX (F/I) was required to see an effect of the cAMP/PKA pathway. By contrast, at lower ANO1 expression levels, which produced current similar to the native ANO1 current, IRBIT similarly increased the current density, and stimulation with 1 μM forskolin and 10 μM IBMX was sufficient to prominently enhance the current. This may be due to a need for an optimal ANO1/cAMP pathway ratio and thus all experiments were at the low ANO1 expression.

In addition, the lower ANO1 expression minimizes the effects of channel multimerization and crowding, resulting in conditions to closely resemble the native situation. We compared the voltage- and time-dependent activation of ANO1 current at multiple Ca^{2+} concentrations and show the results at 10 μM Ca^{2+} in supplementary Fig. 14K. In both case, the current exhibited clear outward rectification and time dependence. Supplementary Fig. 1A shows measurements taken at 0.3 and 30 μM Ca^{2+} with ANO1 expressed at high levels and the linear I/V measured at high Ca^{2+} .

Finally, because these results could be physiologically relevant at the resting membrane potential of cells containing 30-50 mM chloride in their cytoplasm, the authors should perform their electrophysiologic analysis at potential near the resting membrane potential and using a cytosolic solution containing around 40 mM chloride.

Response: As shown in supplementary Fig. 1D reducing pipette Cl^- to 40 had no effect on the response to IRBIT and F/I stimulation.

5. There seems to be a misinterpretation regarding the expression versus localisation of proteins in the TIRF experiments. While the TIRF data show membrane localisation of ANO1, this does not imply a change in its expression levels, as suggested in the manuscript. The author's method said, "The normalization of puncta intensity of the selected cell area was analysed using the NIS-Analysis

software provided by Nikon". How the areas were selected and whether the puncta were individually analysed is unknown. Also, in lane 323, the authors claim, "TIRF measurements showed that inhibition of ANO1 by E-Syt2 was in part due to reduced ANO1 surface expression, even when increased by IRBIT (Figure 5C)". Again, getting the surface expression of ANO1 from the TIRF data is problematic because the experimentalist is only sampling a small area of the membrane. A more punctuated pattern can be observed within the same location or an even smaller amount of protein. However, it is problematic to translate such a piece of information into a global increase or decrease in protein expression. To support that claim of increased expression, a Western blot analysis using membrane protein isolated from biotinylated membranes should be performed to confirm any changes in protein levels. TIRF analysis should include a membrane marker to ensure accurate identification of membrane localisation. Figure 2G shows ANO1 fluorescence data without and with overexpressed IRBIT. How do we know that IRBIT is present in the cells shown in the right panels? The puncta and the whole cell current at +100 mV increased, but such an increase seems less spectacular than the surface increase of ANO1 (panel 2H).

Response: We agree with this comment. To address your concerns, we made the following modifications:

First, we have included additional surface biotinylation assays to show the effects of E-Syt1, E-Syt2 and AC6 with and without F/I stimulation on surface ANO1, and we have correlated the results with their effect on the ANO1 TIRF signal. Once again, there is a perfect correlation between the biotinylation results and the TIRF measurements.

Second, we have added Co-IP measurements involving the E-Syts and ANO1.

Third, we have changed description of the TIRF data, which now states "changes in the level of ANO1 at the TIRF field".

Fourth, we have revised the method section to include the following: "Because many treatments increased both the number of puncta and fluorescence intensity of existing puncta, the change of fluorescence at the TIRF field was analyzed by measuring fluorescence intensity that was then normalized. The normalization of puncta intensity of the selected cell area was analyzed using the NIS-Analysis software provided by Nikon Imaging.

Confocal images were captured using a Yokogawa CSU-X1 with Filter Wheel Control and shutter, a Photometrics® Evolve Delta camera, and Olympus UPlanSApo objectives (60X, 1.35 NA Oil immersion, infinity/0.13-0.19/FN22)".

As indicated in the manuscript, we are fully aware that TIRF microscopy resolution ranges between 100-200 nm and, thus, does not exclusively report on the surface expression of proteins. . **However, it is important to note that TIRF is the best method to get information on surface proteins in live cells that can be best correlated with the measurement of cell function.** To be able to use TIRF for evaluating changes (rather than absolute values) in ANO1 surface expression, we did several correlations between changes in TIRF measurements and biotinylation assays. We believe that the strong correlation between the two methods justifies the use of TIRF, within this limited context, for inferring changes in live cells surface expression as a contributing factor to variations in current density.

The IRBIT protein is tagged with CFP and thus we are certain that the cells express IRBIT.

6. The assumed interaction between TMEM16A and different cytosolic proteins is another critical issue. In some cases, this assumption is supported by the FRET data. However, co-immunoprecipitation assays should be performed to confirm that all those proteins interact within the

native cells. This is particularly relevant because all current experiments used a heterologous expression system that overexpressed the proteins.

Response: FRET provides information on localization within 2-7 nm and interaction in live cells and is the primary technique that is extensively used to assay protein-protein interaction in live cells. To complement the FRET experiments, we added Co-IP experiments to show the Co-IP of ANO1 with E-Syt1 and E-Syt2 in Figs. 4 and 5.

7. Several grammatical errors in the manuscript need to be fixed. Some of these errors are important. For example, in line 131, it is stated that the work was done using the TMEM16A variant a. However, the numbering indicates that the variant used in this work was the ac. For example, residue 673 corresponds to S673 in the ac variant. However, the homologous residue in the variant a is S669. Another example is in line 438; ANO8 and ANO6 were mistyped instead of AC8 and AC6. Also, there are no columns in Figure 3A. Even though some of these errors may have had little incidence in the final interpretation, they can confuse new readers and no experts in the field.

Response: We apologize for these errors and thank you very much for noticing them. These and other similar errors were corrected.

8. The organisation of the figures is unclear, making it difficult to follow the main concepts. Implementing a colour-coded scheme for the graphs would improve the readability and help track the results more quickly. Maintain uniformity in the graph labels by standardising the font size and style across all figures. Keep plot size consistent and balance the distribution of elements within the figures to improve clarity and visual consistency.

Response: Thank you for your suggestions. To the extent possible, we are using a consistent color scheme, particularly for the basic conditions of ANO1±forskolin (black, blue) and ANO1+IRBIT±forskolin (green, red). Regarding the font sizes, we are using the same font but unfortunately the Prism software presents them as different size in some Figures. In these cases, we increase or reduce the font size in order to be as uniform as possible.

We agree that the flow of the presentation needed improvements because of the extensive data provided. As indicated above, a significant number of panels were transferred to the supplementary figures to reduce the panels in the main Figures to the minimum necessary. We hope that this increased the clarity of the results section and improved the presentation.

9. The phosphorylation of residue S673 is another critical issue. Ko et al. showed that residue S673 is the target of CaMKII (<https://doi.org/10.1073/pnas.2014520117>). Accordingly, they showed that KN-62, a blocker of CaMKII, increased TMEM16A current, whereas KT5720, an inhibitor of PKA, had no effect. And yet, in this work, the authors claim that PKA phosphorylates S673. Furthermore, according to Ko's report, the ANO1 S673D current is more minor than WT's; however, in this work, the ANO1 S673D current is more significant than that of the WT current. These issues need to be resolved.

Response: We are very much aware of Ko et. al. manuscript and it is cited multiple times. The lack of effect in inhibition of the BASAL current by inhibition of PKA is in agreement with our observation were a) forskolin stimulation does not affect the current V_{max} and measuring the current at 0.125 μM Ca^{2+} as done by Ko et. al. will unlikely to show a major effect; b) we also used the amply established and specific PKA inhibitor H89 and showed that it reduced the IRBIT+ Forskolin stimulated current only to the level of the IRBIT stimulated current (brown column in the original Figure 2F). Hence, there is no conflict between these observations.

As shown in the attached Figure, we can reproduce the increased current by inhibition of CaMKII with KN-62, as reported by Ko et. al., (red columns). However, we cannot explain why Ko et. al. did not see an increased current by the S673D mutant. In our hands, the S673A variant did not increase the current. Inhibition of a CaMKII effect by the mutation should have resulted in an increase in the basal activity of this variant, but this is not the case. Moreover, both the S673D and S673A mutations eliminated the increase in current by inhibition of CaMKII activity by KN-62. These observations suggest that CaMKII likely affect the phosphorylation of additional S/T residues, as we found for PKA, which complicates dissecting the exact effect of CaMKII on ANO1 current. We use the S673 mutants only in the context of their effect on the VAPA/IRBIT-mediated modulation of the activation of ANO1 by Ca^{2+} . We do, however, point out the different observations in the manuscript and the potential differential regulation of ANO1 by PKA and by CaMKII.

10. Line 155: TMEM16A is regulated but not activated by IRBIT.

Response: Thank you. Activated was changed to regulated.

11. Lines 181-184: How much PIP2 was depleted? How long were the rapamycin stimuli? The current magnitudes are very small!

Response: The Figure legend states, "The cells were treated with vehicle (open columns and symbols) or 0.2 μM rapamycin for 2 min to deplete $\text{PI}(4,5)\text{P}_2$ before current measurements (filled columns and symbols)". Several studies reported nearly complete $\text{PI}(4,5)\text{P}_2$ depletion by this treatment, which we verified in previous studies (PMID: 35416932, 37607230).

12. Fig 5: It seems that E-Syt2 does not inhibit the current. Instead, the Ca sensitivity is affected. The current is smaller because it is less sensitive to Ca.

Response: This is completely correct and is described as such in Figures 5D-F. The strong effect on V_{max} is observed only when cells expressing E-Syt2 are stimulated with forskolin.

13. The Hill Equation cannot describe the data in Fig 5H (Now 5D). It is more complex than that. Therefore, the K_{ms} in Fig 5J (Now 5F) are not credible.

Response: We agree that the Ca^{2+} dependence curves are fairly complicated. The number of Ca^{2+} concentrations used are not sufficient to make the fitting and the K_{m} values absolute and further fittings did not result in improved values. However, the purpose of this Figure is to show that the major effect of E-Syt2 is to regulate the apparent affinity of ANO1 for activation by Ca^{2+} , and this is demonstrated by the results. We modified the text to describe the results in Figure 5D more cautiously.

14. In lane 344, the authors suggest that "it appears that the various combinations of E-Syt2 and IRBIT make complexes with different ACs to differentially determine the surface expression level of ANO1 and its activation by Ca^{2+} ". However, in addition to the author's suggestion, the open probability could also be affected more than the number of channels. Such affection would also help to explain the result and should be discussed at the very least.

Response: Thank you for the comment. We agree that changes in NPo can explain the various effects observed with E-Syt2 and IRBIT and revised the text to account for it, as it reads now

“Although the fittings to obtain the app-Km values were not very tight, they are sufficient to conclude that the primary effect of E-Syt2 is changing the ANO1 apparent affinity for Ca²⁺. In addition, it appears that the various combinations of E-Syt2 and IRBIT form complexes with distinct ACs to differentially determine surface expression level of ANO1, to control ANO1 open probability and its activation by Ca²⁺.”

15. The labels of the columns should be clear because there are multiple treatments. It is confusing to see a “+” before the label because it is not clear what the interpretation should be. Also, if possible, a colour-coded pattern should be followed throughout the manuscript figures.

Response: Thank you. We use the “+” to reduce crowding in the figures and now indicate so in the legend to Figure 2 and indicate that this is done for all other Figures. In response to comment 8, we indicate the color coding that we used throughout.

Reviewer #3 (Remarks to the Author):

The study reveals a new function of STIM1 in the assembly and transduction of the cAMP signaling pathway and elucidate a novel mechanism regulating ANO1 surface expression and Ca²⁺ gating. I have some relatively minor comments.

Response: We greatly appreciate the positive and constructive comments.

INTRODUCTION

1. line 50: "ANO1 is a decision-making channel" - what is a decision-making channel? Clarify or delete

Response: The sentence was revised to read “In epithelial tissues, such as the salivary glands, the pancreas and the airways, ANO1 drives epithelial fluid and electrolyte secretion”

2. Overall, the introduction is far too long - please condense to the essentials

Response: Thank you. As requested, we deleted an entire paragraph and condensed the introduction to the essentials.

METHODS and DATA

1. It is not clear why IRBIT did not increase the current beyond the current measured with ANO1 and VAPA if there is synergy, as the authors say - can you clarify?

Response: The synergistic effects can be seen best in examining Figure 3J-L that requires examining the entire dose response to Ca²⁺. IRBIT increased the apparent affinity for activation by Ca²⁺ measured in the presence of VAPA with minimal effect on V_{max}. Stimulation with PKA in the presence of IRBIT+VAPA now markedly increased the ANO1 current V_{max}.

2. The AC experiments are not clear-AC6 and AC8 are both essential (Lines 292-304) - which goes along with the complexity described in lines 410-436.

Response: Thank you. We agree. It is apparent that the effect of the ACs is determined by the combination of the tethers that recruit them and the junctional localization of the complexes. We clarified this in the manuscript.

3. Line 436: compartmentalization of ACs is not as well studied as that of AKAPs. It should be mentioned here. I am glad the authors studied AKAPs, however, see below

Response: Thank you. The sentence was revised to read "Although not studied as well as the AKAPs, compartmentalization of the AKAPs implies similar compartmentalization of the ACs".

4. It is not clear if the authors are studying PKA-I or -II (R1a or R2a). By the AKAPs, it is PKA-II but it needs to be clarified and experimentally confirmed

Response: Thank you. Indeed, we are studying PKA-II. In the new Supplementary Fig. 10G we used FRET to show that ANO1 primarily interacts with PKA-II in the presence and absence of AKAP5.

Figures

...are hard to see...

Response: Thank you. By moving substantial number of the panels to the supplement, we were able to increase the size of the figures to increase readability.

Remaining reviewers' comments:

1. The authors claim that IRBIT regulates TMEM16A affinity. How can this happen? The Ca affinity is an intrinsic property of TMEM16A that depends on the arrangements of acidic residues forming a pocket holding 2 Ca ions. Enhancing or decreasing the affinity would require altering the pocket structure. How can IRBIT accomplish this affinity change? The authors should indicate in the main text that the change in the apparent K, estimated using the Hill Equation, suggests that the affinity may vary. Also add a line in the methods indicating how the apparent K_d was estimated.

Response: Please note that we indicate in the manuscript that IRBIT regulates the effect of the cAMP/PKA pathway that phosphorylates ANO1 to modulate activation of ANO1 by Ca²⁺. This is based on the results shown in Figures 2A-C (red), 3A (red), 4H-J (blue), while Figure 7A shows that the effects of IRBIT require cAMP generation by adenylyl cyclase 8.

We modified the highlighted text in the last paragraph of page 3 to clarify this, which now reads "IRBIT increased the current V_{max} independent of the cAMP pathway but increased the apparent affinity (app-K_m) for activation of ANO1 by Ca²⁺ that was dependent of the cAMP primarily."

2. I think the authors are overextending their conclusions. In their Discussion (Lines 735-746), the authors described their hypothesis and explained their observations on how phosphorylation would change channel density and apparent affinity. For example, they said "Cell stimulation that releases Ca²⁺ from the ER results in STIM1 translocation along with E-Syt1 to ER/PM junctions. The STIM1 and E-Syt1 translocation increases junctional PI(4)P and PI(4,5)P₂ levels, facilitating formation of the ANO1-VANZ-E-Syt1 complex and its association with the AC8-AKAP5-PKA complex, leading to the phosphorylation of ANO1 Serine 673. As a result, the reserve ANO1 channels close to the PM are inserted into the PM, and ANO1 adopts a sensitized conformation with high Ca²⁺ affinity, becoming fully activated at 1 μM Ca".

a) Such an orderly string of interactions requires time to occur and reach a steady state. It has been previously shown that the chloride current follows the Ca signal with a very short delay (DOI: 10.1113/jphysiol.2001.013453). The onset of the chloride current has a time constant smaller than 20 ms in parotid acinar cells upon a Ca jump. I wonder whether the interactions and channel translocation described here can occur on this time scale to support the proposed hypothesis.

Response: It does not matter what the onset of the Cl⁻ current is because the current does not start until Ca²⁺ rises. GPCR stimulation increases Ca²⁺ in about 100 ms and therefore physiologically the onset of the Cl⁻ current is 100 ms. Thus, the rapidity of activation of the channel by Ca²⁺ is not relevant to our discussion.

b) The so-called reserve of ANO1 channels must be nearly equal to that in the plasma membrane to increase ~2-fold V_{max}. Is there any evidence to back up this idea?

Response: It is not clear to us where the reviewer is taking the numbers from. Figure 2B shows that IRBIT increases the V_{max} from 12.4 to 16.4 pA/pF (32%), which correlates very well with the 36% increase in surface ANO1 in Figure 2E.

c) The apparent affinity of ANO1 has been determined from inside-out patches. Under this experimental condition ANO1 displays high affinity (<https://doi.org/10.1085/jgp.201611650>, doi:10.1371/journal.pone.0086734). The apparent affinity is like that determined from whole acinar cells (see cite above) and higher than those here described. However, this experimental condition does not support the mechanism proposed by the authors. How can the authors' hypothesis be reconciled with those observations?

Response: Again, this is not really a relevant point to our discussion on two levels. It is not possible to compare Ca^{2+} affinity measured in a non-cellular system like excised patches and that measured in intact cells like our measurements. Second, we do not claim that the affinity measured in excised patches is too high or erroneous. Our studies show that ANO1 Ca^{2+} affinity is regulated by the cAMP pathway. This is irrespective of the basal Ca^{2+} affinity measured in any system and take place independent what the basal affinity of the unphosphorylated ANO1.

d) the 5OYB structure, which has 2 Ca bound, have a non-conductive pore because the pore diameter is too narrow to allow chloride permeation. Also, molecular dynamics simulations (<https://doi.org/10.1038/s42003-021-01782-2>) indicate that the pore of the 5OYB structure is not conductive. In this work, it was shown that PIP2 ligation induced pore dilation, which allowed chloride permeation. Therefore, the idea that “binding of a second Ca^{2+} ion, and stabilization of the active conformation” needs to be revised.

Response: Thank you for this comment. The pertinent sentence was revised. Please see highlighted text at the end of the discussion which now reads “The binding of 1 Ca^{2+} ion causes rearrangement of TM6 from an α to π conformation, resulting in increased affinity of the Ca^{2+} binding site, binding of a second Ca^{2+} ion, and perhaps enlargement of the pore by $\text{PI}(4,5)\text{P}_2$ stabilization of the active conformation” with the appropriate references.

3. The data with ATP in the pipette solution should be in the main text because a phosphorylation mechanism seems to be central for explaining the data. Is the S673A mutant insensitive to IRBIT? When ATP is added to the pipette solution to dialyze the cytosol, I doubted that ATP would be compartmentalized. Please clarify what compartmentalized ATP means.

Response: Thank you for this comment. The Figure was transferred to the main Figures and is now Fig. 2F.

It is very well known that glycolysis compartmentalizes some of the ATP, for example to fuel the Na^+/K^+ pump. Searching PubMed for “ Na^+/K^+ ATPase, compartmentalized ATP” yields 35 references and there are many, many more.

The effect of IRBIT on ANO1(S673A) is shown in Fig. 2G (previously 2E), and as expected and stated in the manuscript, the mutant did not reduce the effect of IRBIT on V_{max} but only eliminated the effect of F/I on apparent K_m for Ca^{2+} in activation of ANO1. This has been and is discussed in detail in the manuscript.

4. Please clarify the legend of the Supplementary Figure 2 (F-H), which shows the total ANO1 currents of Figures 2J-L.

Response: Thank you for this comment. A sentence in page 3, the beginning of the last paragraph now state “In this and all other Ca^{2+} -dependence of ANO1 activation, to obtain the Ca^{2+} -dependent current, the current measured in the presence of 5 mM EGTA and no added Ca^{2+} was subtracted. and total current (Supplementary Figs. 1F-H).”.